# Limiting factors for charge generation in low-offset fullerene-based organic solar cells

Anna Jungbluth [1], Eunkyung Cho [2,3], Alberto Privitera [1,4], Kaila M. Yallum [5], Pascal Kaienburg [1], Andreas E. Lauritzen [1], Thomas Derrien [6,7], Sameer V. Kesava[1], Irfan Habib [1], Saied Md Pratik [2], Natalie Banerji[5], Jean-Luc Brédas [2], Veaceslav Coropceanu [2] & Moritz Riede [1] ✉

Free charge generation after photoexcitation of donor or acceptor molecules in organic solar cells generally proceeds via (1) formation of charge transfer states and (2) their dissociation into charge separated states. Research often either focuses on the first component or the combined effect of both processes. Here, we provide evidence that charge transfer state dissociation rather than formation presents a major bottleneck for free charge generation in fullerene-based blends with low energetic offsets between singlet and charge transfer states. We investigate devices based on dilute donor content blends of (fluorinated) ZnPc:$C_{60}$ and perform density functional theory calculations, device characterization, transient absorption spectroscopy and time-resolved electron paramagnetic resonance measurements. We draw a comprehensive picture of how energies and transitions between singlet, charge transfer, and charge separated states change upon ZnPc fluorination. We find that a significant reduction in photocurrent can be attributed to increasingly inefficient charge transfer state dissociation. With this, our work highlights potential reasons why low offset fullerene systems do not show the high performance of non-fullerene acceptors.

Organic solar cells (OSCs) have a very low environmental footprint and are projected to become the most affordable source of solar energy[1]. However, the commercial success of OSCs has long been hindered by their low power conversion efficiencies (PCEs) compared to the detailed balance limit[2]. The main reasons for low PCEs are the large energy losses of 100 s meV that reduce the open-circuit voltage ($V_{OC}$) compared to the optical energy gap ($E_{opt}$) of the light-absorbing molecules[3,4]. These energy losses arise from radiative and non-radiative recombination, often facilitated via the charge transfer (CT) states that form at the donor-acceptor interface[5]. In addition to a reduced $V_{OC}$, free charge generation, which generally proceeds via (1)

the population of CT states after local excitation (LE) of donor or acceptor molecules and (2) CT state dissociation into the charge separated (CS) state, is often also limited. While the efficiency of CT state formation depends on the energetic difference between the molecular optical gap (determined by the lowest energy singlet state of the system) and the CT state energy, i.e., $\triangle E_{CT} = E_{opt} - E_{CT}$, CT state dissociation is influenced by the energetic barrier between the CT and CS states, i.e., $\triangle E_{CS} = E_{CT} - E_{CS}$. This is in addition to other factors influencing transition rates, such as reorganization energies[6] and competition between charge transfer, charge separation, and recombination.

[1]Department of Physics, The University of Oxford, Oxford, Oxfordshire OX13PJ, UK. [2]Department of Chemistry and Biochemistry, The University of Arizona, Tucson, AZ 85721-0088, USA. [3]Division of Energy Technology, DGIST, Daegu 42988, Republic of Korea. [4]Department of Industrial Engineering and INSTM Research Unit, University of Florence, 50139 Firenze, Italy. [5]Department of Chemistry, Biochemistry and Pharmaceutical Sciences, University of Bern, 3012 Bern, Switzerland. [6]Diamond Light Source, Didcot, Oxfordshire OX11 0DE, UK. [7]Present address: Living Systems Institute, University of Exeter, Exeter EX4 4QD, UK. ✉e-mail: moritz.riede@physics.ox.ac.uk

Previous work studying the dynamics of charge separation primarily focused on understanding CT state formation (e.g., refs. 7–9), since reducing $\triangle E_{CT}$ has proven a successful strategy for decreasing voltage losses by reducing the energetic difference between absorbing and emitting states. However, this often comes at the cost of free charge generation. Based on studies of fullerenes, which were the standard acceptor molecules in the OSC research field for decades, it was believed that $\triangle E_{CT}$ of 100 s meV was needed for efficient charge transfer, resulting in an apparent trade-off between high $V_{OC}$ and high photocurrents[10,11]. The emergence of non-fullerene acceptors (NFAs) has brought this tradeoff into question, as OSCs based on NFA blends achieve efficient free charge generation and PCEs up to 19%[12] with very low or vanishing $\triangle E_{CT}$[13–16]. Understanding the factors that lead to efficient free charge generation is still a major focus of current research[4,16–19].

Furthermore, while the CT – CS transition has traditionally received less attention in the literature, recent work demonstrated for polymer:NFA blends that changing the energetics at the donor:acceptor interface not only changes $\triangle E_{CT}$, but also impacts the efficiency of CT state dissociation[18,19]. This challenges the common understanding that donor and acceptor molecular energy levels primarily influence the efficiency of CT state formation, and that $\triangle E_{CT}$ and not $\triangle E_{CS}$ is the limiting factor for efficient free charge generation. Following from these initial insights into the charge separation mechanisms of NFAs, the question remains why low energetic offsets work for some NFAs but generally do not work for fullerenes. Beyond a

scholarly interest in fullerene acceptors, they are highly relevant for upscaling industrial production, since their cost and synthetic complexity are much lower than those of NFAs.

In this work, we investigate how voltage losses, charge transfer, and charge dissociation processes depend on interfacial molecular energetic offsets and electronic coupling in fullerene-based blends. More specifically, we study model systems of dilute bulk heterojunctions (donor:acceptor ratio of 5:95 wt%) of zinc-phthalocyanines (ZnPc) or its fluorinated derivates (F4ZnPc, F8ZnPc, F16ZnPc) as donors and $C_{60}$ as the electron acceptor (Fig. 1a). Current density-voltage (J – V) measurements of the devices show an increase in the $V_{oc}$ between the dilute ZnPc and F16ZnPc blends that can be explained by the fluorination-induced shifts of the molecular energy levels. This change in $V_{OC}$ is accompanied by a strong decrease in charge separation resulting in a severely reduced short-circuit current density ($J_{SC}$). Through density functional theory (DFT) calculations, transient absorption spectroscopy (TAS), and time-resolved electron paramagnetic resonance (trEPR) measurements, we draw a comprehensive picture of how LE, CT, and CS state energies, and the energetic transitions between these states, change upon fluorination of ZnPc. We show that CT state dissociation largely limits free charge generation across our devices and only with extremely small interfacial energetic offsets CT state formation becomes an obstacle. With this our results highlight the need to better understand the CT – CS transition, and to couple transition efficiencies to molecular and blend properties.

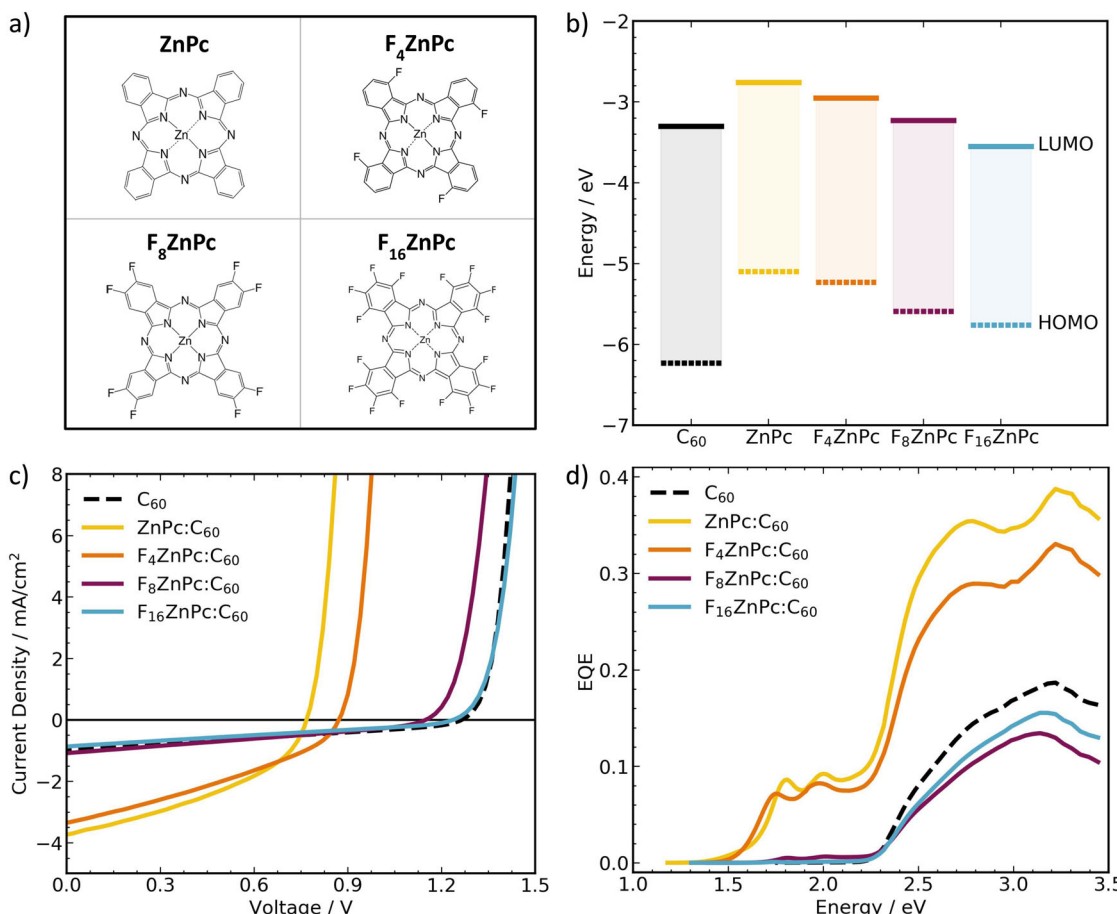

**Fig. 1 | Donor molecular structure and device characteristics of FxZnPc:C60. a** Molecular structures of the investigated donor molecules. **b** Highest occupied molecular orbital (HOMO) and lowest unoccupied molecular orbital (LUMO) energies of the $F_xZnPc$ and $C_{60}$ molecules. **c** Current density-voltage (J – V) characteristics under simulated 1 Sun illumination and **d** external quantum efficiency (EQE) of the best performing (highest PCE) dilute $F_xZnPc:C_{60}$ (5:95 wt%) systems at

room temperature. The performance of a neat $C_{60}$ reference device is shown for comparison. The J – V data was not mismatch corrected. However, due to the strong similarities in the absorption profiles (see Supplementary Figs. S5 and S6), mismatch correction would result in similar scaling for all samples and would preserve the observed performance trends.

## Results

### Device Performance

To investigate the impact of interfacial energetic offsets on charge transfer and separation, we study the performance of dilute OSCs based on $F_xZnPc$ (x = 0, 4, 8, 16) and $C_{60}$ (Fig. 1). Fluorination of ZnPc gradually shifts the HOMO and LUMO energies away from the vacuum level (i.e., increasing both the electron affinity and ionization potential), keeping the molecular singlet energy ($E_{S1}$) roughly constant (Fig. 1b)[20–23]. This shift in donor molecular orbital energies is accompanied by a corresponding shift in the interfacial CT state energy in $F_xZnPc:C_{60}$ bulk heterojunctions (BHJs)[24]. For small-molecule vacuum thermally evaporated OSCs, low donor content BHJs (in our case with donor:acceptor ratios of 5:95 wt%) were shown to produce well-performing devices with reduced recombination losses[25–29]. In addition, grazing-incidence wide-angle X-ray scattering (GIWAXS) of the dilute blends highlights the lack of crystalline $C_{60}$ features, caused by the interspersed donor molecules that effectively disrupt $C_{60}$ packing (Supplementary Fig. S1)[30–32]. This leads to comparable morphologies and makes dilute $F_xZnPc:C_{60}$ blends ideal model systems to study the impact of gradually shifted energy levels on device performance in terms of voltage losses and photocurrent.

Figure 1c shows the current density-voltage (J – V) characteristics of the best performing (highest PCE) devices for nominally identical device architectures. An overview of the performance trends ($V_{oc}$, $J_{sc}$, $FF$, and $PCE$, including sample statistics) is shown in Supplementary Table S1 and Supplementary Fig. S3. Upon fluorination, the $V_{OC}$ increases from 0.78 ± 0.02 V (ZnPc) to 1.21 ± 0.02 V ($F_{16}ZnPc$). This trend agrees with literature reports for intermixed 1:1 blends of $F_xZnPc:C_{60}$[23,24] and $F_xZnPc/C_{60}$ bilayer OSCs[21]. The improvement in $V_{OC}$ results from the fluorination-induced shift of the donor HOMO energies that increase $E_{CT}$ and reduce the energetic offset between the CT state and the donor or acceptor singlets (discussed in more detail later). This change is accompanied by a decreasing $J_{SC}$ from 3.6 ± 0.2 mA cm$^{-2}$ (ZnPc) to 0.8 ± 0.1 mA cm$^{-2}$ ($F_{16}ZnPc$). Interestingly, the $J_{SC}$ of the $F_8ZnPc$ and $F_{16}ZnPc$ blends is comparable to that of neat $C_{60}$. An in-depth discussion of the contributions to the photocurrent can be found in Supplementary Note 2.

Consequently, while the $V_{OC}$ of our fullerene-based OSCs with low energetic offsets match those of well-performing NFAs, we do not observe the same appreciable photocurrents that some low-offset NFAs show[12–16].

In agreement with the J – V behavior, the external quantum efficiency (EQE) of the dilute blends show improved free charge generation for ZnPc and $F_4ZnPc$ compared to neat $C_{60}$ (Fig. 1d). Free charge generation in the dilute $F_8ZnPc$ and $F_{16}ZnPc$ blends, however, is severely limited, as can be seen by the reduced EQE. Both $C_{60}$ and $F_xZnPc$ contribute to photon absorption, with neat $F_xZnPc$ strongly absorbing between 1.6–2.2 eV, and $C_{60}$ strongly absorbing above 2.2 eV (Supplementary Fig. S5b)[33].

Comparing the EQE and absorbance profiles of the dilute blends, efficient free charge generation in the donor absorption region is only observed for the ZnPc and $F_4ZnPc$ blends (Fig. 1d and Supplementary Fig. S7). The characteristic $F_xZnPc$ band is barely visible for $F_8ZnPc$ and cannot be resolved in the linear EQE of $F_{16}ZnPc$. This suggests that the contribution of donor absorption and subsequent electron transfer to $C_{60}$ to the total photocurrent in these blends is low. In addition, the EQE of the $F_8ZnPc$ and $F_{16}ZnPc$ blends in the $C_{60}$ absorption region is roughly equal to that of a neat $C_{60}$ device. This, combined with the overall reduction in the EQE with higher donor fluorinations, suggests that the $F_8ZnPc:C_{60}$ and $F_{16}ZnPc:C_{60}$ interfaces barely contribute to the splitting of singlet excitons into free charges, independent of whether donor or acceptor molecules are excited. Instead, we hypothesize that photocurrents are mostly generated via inter-$C_{60}$ CT states in neat fullerene clusters (without involvement of the donor molecules)[33] which explains the observed "$C_{60}$-like" J – V behavior of the $F_8ZnPc$ and

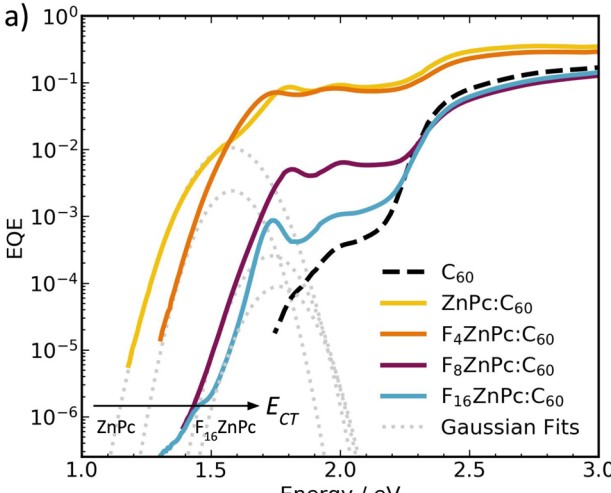

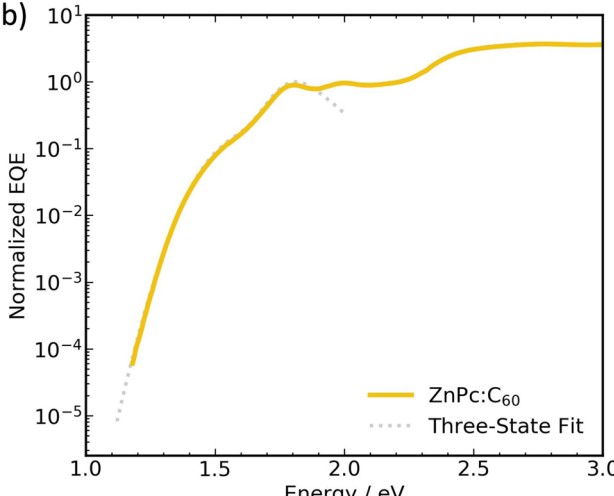

**Fig. 2 | Measured external quantum efficiency (EQE) spectra and spectral fits.** **a** EQE spectra (solid lines), including Gaussian fits of the charge transfer (CT) state (dotted gray lines) for all $F_xZnPc:C_{60}$ blends. A more detailed overview of the fits is shown in Supplementary Fig. S7. **b** Three-state vibronic fit (dotted gray line) of the ZnPc:$C_{60}$ EQE spectrum (solid yellow line). The fitting parameters are listed in Supplementary Table S12.

$F_{16}ZnPc$ blends. This allows us to formally categorize ZnPc/$F_4ZnPc$ as systems with efficient charge separation but $V_{oc}<1V$, and $F_8ZnPc$/$F_{16}ZnPc$ as systems with $V_{oc}>1V$ but limited photocurrents.

### Energy levels

To contextualize the device performances of the fluorinated blends and connect them to molecular and CT state energies, we modeled the CT state absorption bands in the sensitive EQE spectra and performed density functional theory (DFT) calculations for comparison.

Experimental or theoretical characterization of CT states can be challenging, as each method comes with its own particularities[3]. One approach to characterize CT states involves deconvolution of the low-energy tails of EQE spectra through Gaussian functions using classical Marcus theory (Fig. 2a)[5,34]. In brief, this involves fitting the lowest energy singlet state and then subtracting its contribution from the EQE spectrum to fit the remaining sub-gap absorption as the CT state (see Supplementary Fig. S9 and Supplementary Table S2 for more details)[3,35–37]. This yields $E_{CT}$ values of 1.38 ± 0.01 eV, 1.45 ± 0.01 eV, 1.59 ± 0.01 eV, and 1.61 ± 0.01 eV for ZnPc:$C_{60}$, $F_4ZnPc:C_{60}$, $F_8ZnPc:C_{60}$, and $F_{16}ZnPc:C_{60}$, respectively (Table 1). However, looking

**Table 1 | Computationally and experimentally derived energy levels**

|  | ZnPc:$C_{60}$ | $F_4$ZnPc:$C_{60}$ | $F_8$ZnPc:$C_{60}$ | $F_{16}$ZnPc:$C_{60}$ |
|---|---|---|---|---|
| $E_{LE,C_{60}}$ (eV) [DFT] | 2.12 | 2.12 | 2.12 | 2.13 |
| $E_{LE,F_xZnPc}$ (eV) [DFT] | 2.25 | 2.20 | 2.27 | 2.12 |
| $E_{LE,F_xZnPc}$ (eV) [MT] | 1.68 | 1.59 | 1.73 | 1.71 |
| $E_{CS}$ (eV) [DFT] | 1.80 | 1.93 | 2.29 | 2.46 |
| $E_{CT}$ (eV) [TDDFT] | 1.30–1.34 | 1.34–1.39 | 1.47–1.52 | 1.45–1.52 |
| $E_{CT}$ (eV) [MT] | 1.38 | 1.45 | 1.59 | 1.61 |
| $E_{CT}$ (eV) [TSM] | 1.38 | 1.45 | - | - |

The local excitation (LE), charge separated (CS) or charge transfer (CT) state energy levels were determined using density functional theory (DFT) and time dependent DFT calculations, Marcus theory (MT), or three-state vibronic model (TSM) fitting of external quantum efficiency (EQE) spectra. The DFT calculations were performed for isolated $F_x$ZnPc:$C_{60}$ complexes for ε = 4, and using equations 1 and 2. Here, $E_{CT}$ refers to the energy of the singlet CT state, with the triplet CT state energy differing by only a few meV. More information on the EQE fitting using Marcus theory or the three-state vibronic model is included in Supplementary Fig. S9 and Table S10.

at the EQE on a logarithmic scale (Fig. 2a), only the ZnPc:$C_{60}$ system shows a clearly resolvable peak in the sub-gap region, attributable to CT state absorption. With increasing donor fluorination, the CT state shoulder merges with the LE singlet transitions. This makes a simple spectral band deconvolution approach ambiguous. Moreover, two-state models (like classical Marcus theory) only account for coupling between the CT and ground state (GS), even though the intensity and shape of the CT absorption band also depends on the hybridization between CT and LE states[38,39].

In this context, three-state dynamic vibronic models are generally needed[38,39]. The simulations of the CT state absorption bands of the ZnPc and $F_4$ZnPc blends using a three-state model show that a good fit of the EQE can only be obtained when accounting for the hybridization between CT and LE states (see Fig. 2b, Supplementary Fig. S16 and Supplementary Table S12 for details). The CT state energies and $\triangle E_{CT}$ derived from three-state vibronic fitting are 1.38 eV and 0.3 eV for ZnPc:$C_{60}$, and 1.45 eV and 0.2 eV for $F_4$ZnPc:$C_{60}$, respectively (Table 1). These values are consistent with those predicted using Marcus theory. In the case of $F_8$ZnPc:$C_{60}$ and $F_{16}$ZnPc:$C_{60}$, we were unable to simulate the low-energy EQE bands with a meaningful and unambiguous choice of microscopic parameters. Here, we note that the classical and dynamic vibronic models used above are designed to interpret CT state absorption bands, not necessarily EQE spectra, which depend on both absorption and the charge separation efficiency of CT and LE states. If the charge separation efficiency is for instance energy-dependent, then EQE fitting to derive electronic-structure information might not work properly. As discussed earlier, while the absorbance of the dilute blends is constant across donor fluorinations (Supplementary Fig. S6), the EQE reduces for $F_8$ZnPc and $F_{16}$ZnPc, with the starkest decrease observable in the donor absorption region (Fig. 2a). This results in a lower internal quantum efficiency (IQE) indicative of reduced donor exciton dissociation in the $F_8$ZnPc and $F_{16}$ZnPc blends, and manifests in what would be observed as an extremely low oscillator strength when fitting the three-state vibronic model.

To validate our estimates of $E_{CT}$, and determine singlet, triplet ($T_1$), and CS state energies, which cannot be easily measured experimentally, we performed DFT and time dependent DFT calculations (discussed in detail in the Supplementary Information). The TDDFT results and $E_{LUMO}/E_{HOMO}$ values are included in Supplementary Tables S4–S8 and were calculated for $C_{60}$ and $F_x$ZnPc molecules embedded into a dielectric medium with ε = 4. The resulting energies of LE, CS, and CT states, including the values derived from EQE fitting, are collected in Table 1 and discussed in the following.

In line with experimental measurements[20–23], our calculations show that fluorination of ZnPc only marginally changes the donor singlet energy, with a mere 0.1 eV decrease between ZnPc and $F_{16}$ZnPc (see Supplementary Tables S4–S7). To estimate $E_{CT}$, we use a simplified but widely employed model[40–42] that approximates the CT state energy as the sum of $E_{CS}$ and the hole-electron electrostatic interaction. This approach seems to overestimate $E_{CT}$ and is only discussed in the Supplementary Information. The TDDFT derived CT state energies span a range from 1.3 eV (ZnPc:$C_{60}$) to 1.5 eV ($F_{16}$ZnPc:$C_{60}$) and are about 0.1 eV smaller than those derived from EQE fitting. Our calculations also confirm that the singlet and triplet CT states in the $F_x$ZnPc:$C_{60}$ blends are quasi-degenerate[43,44], with an energy splitting of 40–70 meV. In agreement with experimental reports[24,45,46], our DFT calculations show that the $F_x$ZnPc triplet is the lowest excited state in the blends at 1.2 eV. For comparison, the $T_1$ of $C_{60}$ is ~1.8 eV. In summary, while the CT states of ZnPc:$C_{60}$ and $F_4$ZnPc:$C_{60}$ are located between the $T_1$ of the donor and acceptor, the CT states of $F_8$ZnPc:$C_{60}$ and $F_{16}$ZnPc:$C_{60}$ are in resonance or above the $T_1$ of both molecules. This has direct implications for geminate recombination pathways as we discuss in the next section.

## Charge separation and recombination

We now relate our characterization of molecular and blend energy levels to charge separation and recombination dynamics, measured via time-resolved electron paramagnetic resonance (trEPR) spectroscopy and transient absorption spectroscopy (TAS). trEPR can probe the dynamics of charge species containing unpaired electron spins, e.g., CT and triplet states, via distinctive spectral signatures with 100 s ns time resolution[47,48]. Through deconvolution of the characteristic polarization patterns resulting from non-Boltzmann population of the triplet sublevels[49–51], trEPR can provide information on the location of triplet states via the zero-field splitting (ZFS) parameters, and distinguish whether these triplets are formed through geminate back electron transfer (BET; spin flip in the CT state and subsequent transfer to donor/acceptor triplets) or inter-system crossing (ISC; spin flip in LE singlet state). While non-geminate recombination can usually not be observed via trEPR due to the lack of spin-polarization, the spin-statistical recombination of uncorrelated charges usually results in a similar population of triplet sublevels[49,51,52]. TAS is sensitive to fast non-geminate recombination, allowing us to obtain a comprehensive picture of the charge dynamics in our studied blends.

In this context, understanding the competition of CT state formation (LE – CT transition) and CT state dissociation (CT – CS transition) with different recombination pathways is key. trEPR is an ideal technique to qualitatively study this competition[52], as ISC from singlet to triplet states competes with CT state formation, and BET from CT to triplet states competes with CT state dissociation. Observing a strong signal of triplet formation via ISC therefore suggests that CT state formation is inefficient, while a strong signal of BET suggests that CT state dissociation is inefficient.

We carried out trEPR measurements at 80 K and using 532 nm laser excitation on the dilute blends (Fig. 3 and Supplementary Figs. S20–S21), and, for comparison, on the neat donors (Supplementary Fig. S19). This excitation wavelength mainly excites $C_{60}$ clusters in the dilute blends, with a smaller contribution from donor excitation. All spectra show two main species: (1) a spectrally narrow signal centered at ~ 345 mT, assigned to photogenerated charges[47,48,53,54] and (2) a broad signal ranging from about 320 – 370 mT, attributed to triplet states generated via BET or ISC[30,49,52]. Looking at Fig. 3 (and Supplementary Figs. S19–S21), the low intensities of the signal of photogenerated charges can be rationalized by quick charge recombination (faster than 100 s ns, which is the time resolution of our trEPR setup)[30,55]. As for the triplet signals, we performed best-fit simulations of the trEPR spectra and report the results in Table 2 for the dilute blends and Supplementary Table S13 for the neat films.

We find that the $T_1$ ZFS parameters of all dilute $F_xZnPc$ blends are comparable, but slightly larger than those of the neat $F_xZnPc$ films. This suggests that triplets in the blends are localized on the donor molecules. No evidence of the $C_{60}$ triplet is observed in the spectra. We hypothesize that triplets generated via ISC on the $C_{60}$ molecules either undergo (1) charge transfer to the CT triplet (if energetically possible), or (2) energy transfer to the donor triplet.

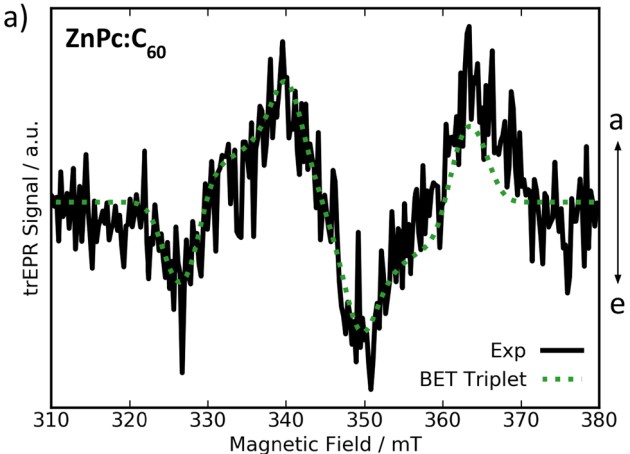

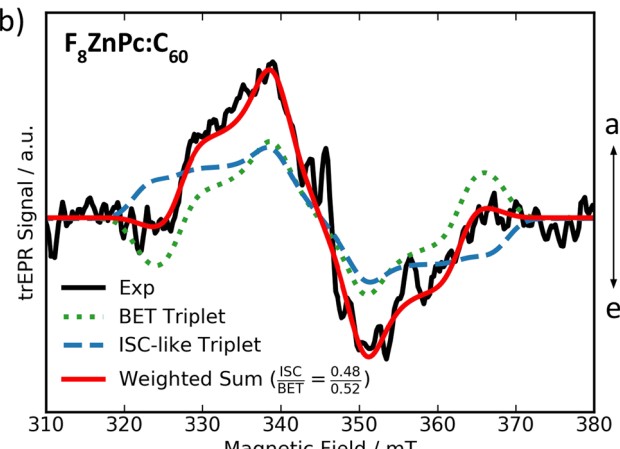

**Fig. 3 | Time-resolved electron paramagnetic resonance (trEPR) spectra and spectral fits.** Measurements were performed at 80 K for the dilute (**a**) ZnPc:$C_{60}$, and (**b**) $F_8ZnPc:C_{60}$ blends. The spectra and fits of the $F_4ZnPc:C_{60}$ and $F_{16}ZnPc:C_{60}$ blends are reported in Supplementary Fig. S20. Black line: trEPR spectra of the dilute blends recorded at 1 μs (integration window 0.8–1.2 μs) after a 532 nm laser pulse. Absorption (*a*) is up and emission (*e*) is down. Red line: best-fit spectral simulations of donor triplets obtained as the sum of two contributions: (1) an intersystem crossing (ISC)-like contribution (blue dashed line) and (2) geminate back electron transfer (BET) to a low-lying $T_1$ triplet state (dotted green line). The best-fit values are reported in Table 2.

Further investigating the $T_1$ population pathways, we only observe BET triplet formation for ZnPc:$C_{60}$ (see Table 2). The strong ISC signal observed for neat ZnPc (discussed in Supplementary Note 6) is quenched for ZnPc:$C_{60}$ blends, suggesting that ISC from an excited singlet state is hindered by fast CT state formation[30]. The weak BET signal with low signal-to-noise ratio in Fig. 3a suggests that geminate BET in ZnPc:$C_{60}$ blends occurs with low probability due to efficient CT state dissociation, rationalizing the high photocurrents observed for this material system (Fig. 1).

In contrast, the fluorinated donor blends show two main contributions to $T_1$ formation: the first resembles the ISC contribution observed in neat $F_xZnPc$ films but with different $T_1$ sublevel populations (for simplicity, we label this process as "ISC-like"), and the second contribution is geminate BET (see Table 2). Due to the weak donor excitation at 532 nm, direct ISC on the donor molecules contributes little to the ISC-like contribution, especially since the donor molecules are surrounded by $C_{60}$ molecules in the dilute blends and should undergo efficient charge transfer. Therefore, the ISC-like contribution likely arises from triplet-triplet energy transfer from $C_{60}$ to $F_xZnPc$, as previously observed in literature[30]. This explains the change in the spin populations of the ISC-like triplet in the dilute blends with respect to the neat films, which suggests a different triplet population mechanism. Due to the low energetic offset between the $S_1$ state of $F_8ZnPc$/$F_{16}ZnPc$ and the CT state of the respective dilute donor blends, it is also possible that the CT and $S_1$ states exist in equilibrium, opening the CT → $S_1(F_{8/16}ZnPc)$ → $T_1$/GS pathway[17,56,57]. Conversely, geminate BET (CT → $T_1$) can be rationalized considering that the donor triplets are the lowest energetic states in the blends, as discussed earlier. As a result, faster BET to the donor triplet than CT state dissociation into the CS state can occur. In the case of $F_{16}ZnPc:C_{60}$, spectral fitting indicates only a weak contribution of triplet formation via BET, and a decent fit can be achieved by neglecting BET altogether (see Supplementary Fig. S21). Instead, the $F_{16}ZnPc$ triplets are mainly populated via the ISC-like contribution of triplet-triplet energy transfer. This observation points to inefficient CT state population.

To expand our investigation of charge dynamics, we performed TAS measurements at 530 nm excitation. TAS complements trEPR well, due to its increased temporal resolution (0.2–1500 ps compared to 0.2–1.2 μs for trEPR) and sensitivity to both geminate and non-geminate charge pairs[49,51]. Figure 4 shows the TAS spectra for ZnPc:$C_{60}$ and $F_8ZnPc:C_{60}$ (measurements for the other blends and neat donors are included in Supplementary Figs. S22–S23).

TAS spectra of the blends at 0.2 ps (red solid lines), show the excited state absorption (ESA) of $C_{60}$ (see inset for comparison to neat $C_{60}$), since initial excitation at 530 nm strongly excites $C_{60}$ clusters (Supplementary Fig S6). Our previous work has shown that for thin films without applied bias, inter-$C_{60}$ CT excitons in $C_{60}$ clusters rapidly (<0.2 ps) localize to Frenkel excitons on isolated $C_{60}$ molecules, leading to the $C_{60}$ ESA observed here[33]. TAS spectra of ZnPc:$C_{60}$ at 0.2 ps also show donor ground state bleaching (GSB) and the corresponding broad ESA at 990 nm, due to partial excitation of ZnPc at 530 nm (see Supplementary Fig S23). For the $F_xZnPc:C_{60}$ blends, direct excitation

**Table 2 | Best fit values obtained from fitting the time-resolved electron paramagnetic resonance spectra of the dilute $F_xZnPc:C_{60}$ blends**

| | Triplet Species | [D E] (MHz) | [$p_x$ $p_y$ $p_z$] | $LW_{ISC}$ (mT) | [$p_{-1}$ $p_o$ $p_{+1}$] | $LW_{BET}$ (mT) | $weight_{ISC}$: $weight_{BET}$ |
|---|---|---|---|---|---|---|---|
| ZnPc:$C_{60}$ | BET | [583 107] | | | [1 0 1] | 3.7 | 0:1 |
| $F_4ZnPc:C_{60}$ | ISC + BET | [646 108] | [0.38 0.42 0.20] | 2.2 | [1 0 1] | 2.3 | 0.46:0.54 |
| $F_8ZnPc:C_{60}$ | ISC + BET | [670 112] | [0.05 0.25 0.70] | 4 | [1 0 1] | 3.9 | 0.48:0.52 |
| $F_{16}ZnPc:C_{60}$ | ISC + BET | [592 115] | [0.14 0.28 0.58] | 3.7 | [1 0 1] | 2.6 | 0.79:0.21 |

For each film, the populating mechanism of the triplet states and the zero-field splitting (ZFS) parameters ([D E]) and the triplet sublevel populations [$p_x$ $p_y$ $p_z$] are reported. Only Gaussian broadening was considered for all triplets to avoid over-parameterizing the fits. The relative weight of intersystem crossing (ISC) and back electron transfer (BET) triplets is reported in the last column.

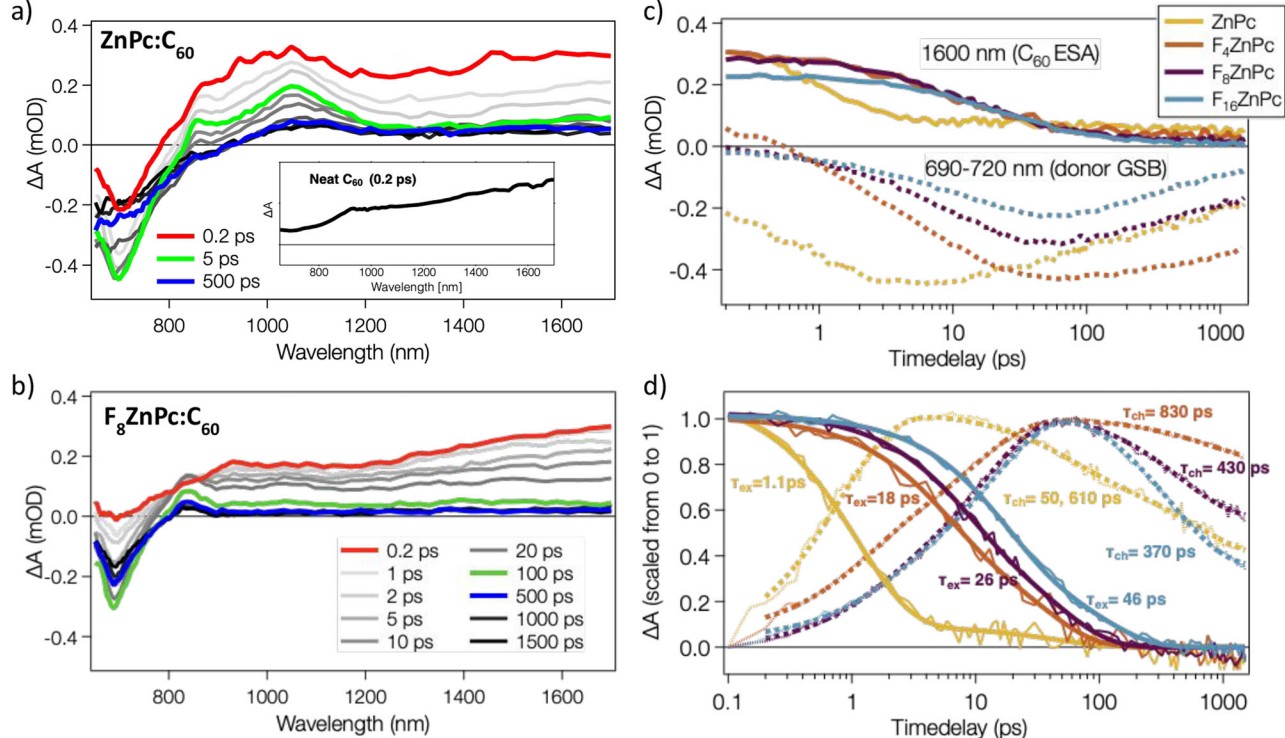

**Fig. 4 | Transient absorption (TA) spectra and dynamics.** The TA spectra were measured at selected time delays after excitation at 530 nm for (**a**) ZnPc:C$_{60}$ and (**b**) F$_8$ZnPc:C$_{60}$ dilute donor blends. TA measurements of F$_4$ZnPc:C$_{60}$ and F$_{16}$ZnPc:C$_{60}$ are included in Supplementary Fig. S22. The inset shows the early spectrum of a neat C$_{60}$ film for reference. **c** Dynamics at 1600 nm (mainly C$_{60}$ excited state absorption (ESA), smooth lines) and at 690–720 nm (maximum of the donor ground state bleaching (GSB), dotted lines) for the four investigated F$_x$ZnPc:C$_{60}$ blends. **d** The same dynamics are scaled between 0 and 1 for better comparison. Thick smooth solid lines are multi-exponential fits to the dynamics at 1600 nm, while thick dotted lines are fits to the dynamics at 690–720 nm. $\tau_{ex}$ and $\tau_{ch}$ show the average timescales for the exciton and charge decay dynamics respectively.

of the donors is less pronounced and only leads to a weak donor GSB, which spectrally overlaps with the dominant C$_{60}$ ESA.

For all blends, the TAS spectra show an increase in the GSB of the donors with time and a concomitant decay of the C$_{60}$ ESA, leading to new spectral signatures encompassing the donor GSB, a sharp peak at 840–860 nm, and a broad band around 1050 nm. This is shown in green in Fig. 4 and Supplementary Fig S22, for the time delay where the donor GSB is maximal (5 ps for ZnPc:C$_{60}$, 100 ps for the F$_x$ZnPc:C$_{60}$ blends). This increase in GSB is indicative of charge and/or energy transfer from excited C$_{60}$ to the donor molecules. Due to the low photoluminescence quantum yield of C$_{60}$, we assume that singlet energy transfer from C$_{60}$ to the donors is minimal and hypothesize that the new spectral signatures primarily stem from charges (donor cations and C$_{60}$ anions). The ZnPc spectral component matches published spectra of ZnPc charges obtained by electrochemical doping and photoexcitation of ZnPc:C$_{60}$ blends[58].

Figure 4c compares the dynamics of the donor GSB (at 690–720 nm) and the C$_{60}$ ESA (at 1600 nm) of all blends, and Fig. 4d shows the scaled dynamics for better comparison. Multi-exponential fits of the exciton dynamics reveal that charge transfer occurs with an average time constant of around 1.1 ps for ZnPc:C$_{60}$ and 18–46 ps for F$_x$ZnPc:C$_{60}$, increasing with donor fluorination. Our previous studies of other dilute donor:C$_{60}$ blends also revealed charge transfer times of a few ps[31], which encompasses C$_{60}$ exciton diffusion to a donor molecule, followed by hole transfer[59]. We rationalize the slower charge transfer in the fluorinated blends with the closer energetic matching between the C$_{60}$ LE singlet and CT states, reducing the driving force and slowing down hole transfer.

Although all blends were pumped at a similar excitation density, the maximal donor GSB reaches a higher amplitude for ZnPc:C$_{60}$ and significantly decreases with increasing fluorination. This indicates that

the efficiency of charge transfer is gradually reduced in the F$_x$ZnPc:C$_{60}$ blends. As a result, charge transfer in the fluorinated blends competes with recombination to the ground or triplet states, which occurs in about 150 ps[33]. Since the triplet state of the donors is the lowest energetic state in the blends, the C$_{60}$ triplet undergoes energy transfer to the F$_x$ZnPc triplet, in agreement with the ISC-like triplet contribution observed via trEPR.

Looking back at the EQE spectra (Fig. 1d), interfacial CT states separate in the ZnPc:C$_{60}$ and F$_4$ZnPc:C$_{60}$ blends, but not in the two systems with higher fluorination. For the time scale relevant to TAS (<1.5 ns), holes are trapped on isolated donor molecules in the dilute blends and charge separation occurs via electrons that delocalize into fullerene clusters[28,31].

While CT state dissociation competes with geminate recombination to either the donor triplet or ground state, charges in the CS state can also recombine non-geminately. In this context, we have previously identified a monomolecular trap-based non-geminate recombination mechanism, whereby free electrons recombine with immobile holes in dilute donor films[31]. Both geminate and non-geminate recombination mechanisms lead to significant charge recombination in the ZnPc:C$_{60}$ blend, with time constants of 50 ps and 610 ps (Fig. 4d), indicative of bound and separated charges. We note, however, that this trap-based recombination is not relevant in devices, where free electrons are rapidly extracted.

The late TAS spectra of the ZnPc:C$_{60}$ blend (e.g., blue solid line at 500 ps in Fig. 4a) have distinct spectral signatures with a broadened donor GSB and red-shifted band at 1080 nm. We assign the latter to the ZnPc triplet state populated via BET, in agreement with our trEPR analysis.

For the F$_4$ZnPc:C$_{60}$ blend, charge recombination is significantly reduced (830 ps time constant) and both the charge and triplet signals

persist at long times (Supplementary Fig. S22). We rationalize this with slower CT state recombination due to the higher $E_{CT}$ of this blend, in line with the energy gap law[60].

For the $F_8ZnPc$ and $F_{16}ZnPc$ blends, on the other hand, recombination accelerates (430 ps and 370 ps, respectively), resulting in inefficient free charge generation. Since BET triplets are less evident in the trEPR of these blends, we hypothesize that CT excitons can transfer to the energetically close donor $S_1$ state, forming Frenkel excitons which then undergo ISC or ground state recombination[17,56,57]. Since all excited states of the donors (i.e., singlet and triplet excitons, bound and separated charges) lead to the GSB signal, a deconvolution of the different spectral contributions in the TAS spectra is not trivial. However, we observe no significant spectral changes for the $F_8ZnPc$ or $F_{16}ZnPc$ blends at late times (e.g., blue solid line at 500 ps in Fig. 4b), suggesting that the 'ISC-like' triplets are generated by a different mechanism than the BET-triplets in the $ZnPc:C_{60}$ and $F_4ZnPc:C_{60}$ blends, outside the TAS time window.

## Discussion

In summary, we studied low donor content blends of $F_xZnPc$ (x = 0, 4, 8, 16) with $C_{60}$ as model systems to investigate the effect of interfacial energetic offsets on charge transfer, charge separation and recombination. Our experimental data and theoretical results show that the observed trends in device performance can be related to changes in the relative positions of LE and CT state energy levels upon donor fluorination. In addition, the transition between CT and CS states needs to be considered to obtain a full picture of charge dynamics in the blends.

In comparison to neat $C_{60}$ films, where free charges are predominantly formed via dissociation of high-energy inter-$C_{60}$ CT states, free charge generation in donor:acceptor blends depends on the formation and dissociation of interfacial CT states. Based on our characterization of molecular and blend energy levels, and charge dynamics measured via trEPR and TAS, we can draw energy level diagrams and highlight the dominant transition pathways observed in the different $F_xZnPc:C_{60}$ blends (Fig. 5 and Supplementary Fig. S24). In the case of $ZnPc:C_{60}$, trEPR and TAS confirm that geminate and non-geminate triplet formation are minimal, indicating that CT state formation and dissociation are efficient, and rationalizing the high photocurrent of this blend. With increasing donor fluorination, charge transfer slows down. For $F_4ZnPc:C_{60}$, charge transfer occurs with an average time constant of 18 ps, significantly slower than the 1.1 ps determined for $ZnPc:C_{60}$. As a result, charge transfer competes with recombination to the ground and triplet states, as evidenced by the emergence of triplet state population via the $S_1(C_{60}) \rightarrow T_1(C_{60}) \rightarrow T_1(F_4ZnPc)$ pathway. Nonetheless, the $F_4ZnPc$ blend shows a similar photocurrent to $ZnPc:C_{60}$, highlighting that slower CT formation is not obstructing free charge generation in this material system.

In the case of $F_8ZnPc:C_{60}$, we observe similar proportions of triplet formation via ISC and BET as for $F_4ZnPc:C_{60}$, with slightly slower charge transfer within 26 ps. In addition, our TAS measurements suggest that CT excitons can transfer back to the energetically close donor $S_1$ state, a transition pathway that is only competitive if CT state dissociation is slow. As a result, we hypothesize that CT state dissociation, rather than CT state formation is limiting free charge generation in the $F_8ZnPc:C_{60}$ blend. The drastic difference in photocurrents between the $F_4ZnPc$ and $F_8ZnPc$ blends can thus be attributed to a decreasing CT − CS transition efficiency. While most CT excitons in $F_4ZnPc:C_{60}$ proceed to the CS state, resulting in a high photocurrent, CT states in $F_8ZnPc:C_{60}$ do not yield free charge carriers and instead, seem to recombine to the ground state. In the case of $F_{16}ZnPc:C_{60}$, both CT state dissociation and CT state formation are limited, as observed via the increasing contribution of ISC that follows from severely slowed down charge transfer.

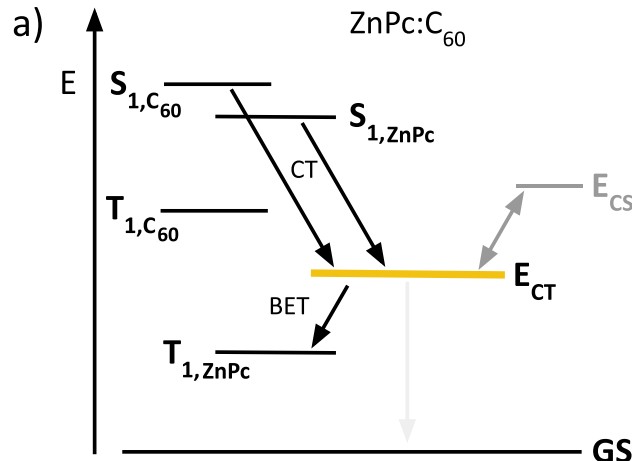

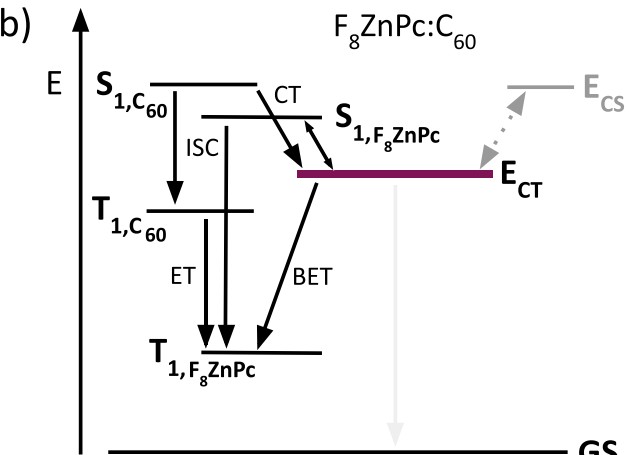

**Fig. 5 | Schematic representations of computationally and experimentally derived energy levels and dominant transition pathways.** The charge transfer (CT), intersystem crossing (ISC), back electron transfer (BET), and energy transfer (ET) processes are shown for dilute (**a**) ZnPc, and (**b**) $F_8ZnPc$ blends with $C_{60}$. The dotted lines signal pathways that we observe with lower probability. The represented CT state is comprised of nearly degenerate singlet and triplet CT states. Although not shown in the diagram, all excited states can directly recombine to the ground state (GS).

The difference in the CT dissociation efficiency of the blends might be due to energetics, i.e., larger $\triangle E_{CS}$, or due to other coupling parameters involved in the CT − CS transition. Note that we assumed $\triangle E_{CS}$ constant in our DFT calculations, which was suitable for estimating how CT state energies change with donor fluorination. For CT dissociation, variations in $\triangle E_{CS}$ become more important. While we are not aware of any methods that can directly measure or accurately calculate $\triangle E_{CS}$, our findings shed light onto the working mechanisms of low energetic offset systems: Many NFA blends show similarly low driving forces as in our studied blends, which is often perceived as paradoxical since free charge generation is still high in these systems. Our results show that the key to understanding the performance of low offset systems, and specifically well performing NFAs, involves disentangling the influence of $\triangle E_{CT}$ and $\triangle E_{CS}$ on free charge generation, as the uniqueness of NFAs might be explained by efficient CT − CS transitions.

## Methods
### Sample fabrication details
All samples were fabricated via vacuum thermal evaporation using an evaporation chamber (Creaphys, EVAP300, base pressure $10^{-7}$ mbar). Prior to deposition, the substrates were sonicated for 10 min in a

solution of 2.5% Hellmanex in DI water, followed by DI water, acetone, and isopropanol. The substrates were treated with $O_2$ plasma for 10 min before being loaded into the evaporation chamber. Full solar cells were fabricated on pre-patterned ITO-coated glass (Eagle XG glass, 20 Ohms/sq, rms roughness <7 Å) and through subsequent evaporation of $MoO_x$ (3 nm, 0.05–0.1 Å/s), $F_xZnPc:C_{60}$ (5:95 wt% ratio, 50 nm, 0.4 Å/s total rate), BPhen (8 nm, 0.1–0.2 Å/s), and Al (-100 nm, 1–2 Å/s). The ITO-coated glass was purchased from Thin Film Devices TFD Inc., USA. Optoelectronic grade $C_{60}$ was purchased from Creaphys GmbH, Germany. ZnPc, $F_4ZnPc$, $F_8ZnPc$, $F_{16}ZnPc$, BPhen, and $MoO_x$ were purchased from Luminescence Technology Corp. The area of the measured device pixels was 0.08 $cm^2$, as determined via the geometric overlap of the patterned ITO and the evaporated layers. Thin film samples of the active layer were fabricated on microscope cover glass for EPR, and fused quartz substrates (type WHQ from Knight Optical Ltd) for absorbance and GIWAXS measurements. The substrates were held at room temperature during the layer-by-layer deposition. Post deposition, all samples were transferred into a $N_2$-filled glovebox without air exposure and encapsulated for further characterization.

## Current density-voltage measurements

Current density-voltage characteristics were performed under simulated AM1.5 g light with 100 $mW/cm^2$ intensity using a sun simulator (Abet Technologies, Sun 2000, Class AAB) and a source meter (Keithley, 2400 Source Measure Unit). The light source was calibrated using an NREL-certified KG5 filtered silicon reference diode. Measurements were performed in both forward and reverse scan direction and in 0.02 V steps. All samples were measured at room temperature.

## External quantum efficiency measurements

Sensitive external quantum efficiency (EQE) measurements were performed using a custom-built setup. White light from a tungsten-halogen light source (Princeton Instruments, TS-428, 250 W) was diffracted by wavelength using a monochromator (Princeton Instruments, Spectra-Pro HRS300, Triple Grating Imaging Spectrograph). Using spectral filters (Thorlabs, edge pass and long pass filters), stray light and higher-order diffractions were removed. The light was modulated using a chopper wheel (Stanford Research Systems, SR450, Optical Chopper) before being focused onto the device under testing. The resulting photocurrent was pre-amplified (Zürich Instruments, HF2TA Current Amplifier) before being read out by a Lock-In amplifier (Zürich Instruments, HF2LI Lock-In Amplifier). The EQE spectra were calculated via calibrated silicon (Thorlabs, FDS100-CAL) and InGaAs (Thorlabs, FGA21-CAL) photodiodes. Temperature-dependent EQE measurements were performed by mounting the sample in a cryostat (Linkam, LTS420 Stage).

## Absorbance measurements

Absorbance measurements were carried out on thin films on quartz using a UV-Vis-NIR photo-spectrometer (Perkin-Elmer, Lambda 1050). Measurements were performed in transmission mode and after collection of a light and dark baseline. The light baseline was performed by measuring the transmission of a clean quartz substrate.

## Ellipsometry measurements

Ellipsometry measurements were carried out using a spectroscopic ellipsometer (J.A. Woollam, RC2 Spectroscopic Ellipsometer) at 55°, 65°, and 75° angles of incidence. The measurements were performed on neat films and donor:acceptor blends. From the measurements of $\Psi$ and $\triangle$, the optical constants, i.e., the refractive index $\eta$ and the extinction coefficient $\kappa$, were determined via optical modeling (J.A. Woollam, CompleteEASE Software).

## GIWAXS experimental details

Grazing incidence wide-angle X-ray scattering (GIWAXS) for all except one sample was carried out at the Surface and Interface Diffraction

beamline (I07) at the Diamond Light Source (DLS) using a beam energy of 20 keV (0.62 Å) and a Pilatus 2 M area detector. The sample-to-detector distance was calibrated using silver behenate (AgBeh) and determined to be 664.8 mm. GIWAXS measurements of the dilute $F_4ZnPc:C_{60}$ blend were carried out at the European Synchrotron Radiation Facility (ESRF) at BM28 (XMAS/THE UK CRG). An X-ray energy of 20 keV (0.62 Å) was used for illumination and 2D images were recorded using a Pilatus 2 M detector. A flight tube with Kapton windows was used to reduce air scattering and the sample-to-detector distance was determined to be 421.6 mm using AgBeh. Images were converted to 2D reciprocal space using the DAWN software package with an applied polarization and solid angle correction[61]. Reciprocal space maps are shown using a logarithmic color scale with limits chosen to facilitate the readers' understanding of scattering features. The positions of the primary out-of-plane peaks were determined from conical slices between 2° and 5° of the out-of-plane axial orientation which were fitted as Lorentzian functions with a linearly varying background. The angular off-set is due to the missing wedge which arises from to the conversion to reciprocal space. The position of the in-plane $\pi - \pi$ stacking peaks were determined from a rectangular region around the Yoneda band (roughly $q_z = 0.05$ $Å^{-1}$) and fitted in the same way. The corresponding distance in real space was found by $d = 2\pi/q$. The position of peaks of the low-donor samples were not fit as their scattering contribution is primarily positioned in the missing wedge making accurate characterization difficult.

## Computational methodology

To determine the electronic structures of $F_xZnPc:C_{60}$ complexes, we considered face-on configurations of $F_xZnPc$ and $C_{60}$ molecules, since previous calculations[62,63] show that this configuration is the most stable. The electronic structure calculations were performed at the density functional theory (DFT) level using the screened range-separated hybrid functional LC-ωhPBE with 20% Hartree-Fock exchange and the 6−31 G** basis set. The GD3BJ dispersion correction was included in all calculations. The range-separation parameters (ω) was optimized for each complex. The excited-state properties were obtained at the time-dependent DFT (TDDFT) level within the Tamm-Dancoff approximation (TDA-TDDFT), also using the screened range-separated hybrid (SRSH) LC-ωhPBE functional with the 6−31 G** basis set[64]. Natural transition orbital (NTO) analyses were carried out to characterize the nature of the excited states. The relaxation energies of the $F_xZnPc$ and $C_{60}$ molecules, related to the decay of CT states to the ground state, were obtained from both the adiabatic potential energy surfaces and normal mode calculations of the neutral and charged states[65]. The effect of electrostatic screening and electronic polarization of the crystal environment (characterized by a dielectric constant) on the frontier orbitals and excited-state energies was considered by modifying the LC-ωhPBE functional according to established literature procedure[40,66]. All DFT/TDDFT calculations were performed by with the Gaussian 16 program[67].

The electronic couplings between the CT and ground states, and between the CT and LE states were computed from the generalized Mulliken–Hush (GMH) method[68] as implemented in the Q-Chem 5.3 package[69].

The zero-field splitting (ZFS) parameters, D and E, were calculated by performing additional single-point calculations on optimized geometries. The calculations were carried out at different levels of theory, including B3LYP/6-31 G**, PBE0/6-31 G**, B3LYP/def2-TZVP, and B3LYP with the def2-TZVP basis set for zinc and the EPR-II basis set for all other atoms. The spin–spin term was calculated based on the UNO determinant[70]. These calculations were carried out with ORCA 5.0.3[71,72].

## EPR experimental details

EPR samples were prepared as thin films of 50 nm thickness deposited on microscope cover glass, cut to a width of 3 mm with a

diamond-tipped glass cutter. The strips were placed in quartz EPR tubes which were sealed in a nitrogen glovebox with a bi-component resin (Devcon 5-Minute Epoxy), such that all EPR measurements were performed without air exposure. All trEPR spectra were recorded on a Bruker Elexsys E580 X-band spectrometer, equipped with dielectric ring resonator (ER 4118X-MD5). The sample temperature was maintained using a nitrogen gas-flow cryostat (Oxford Instruments CF935O) and controlled with an Oxford Instruments ITC503 temperature controller. Laser pulses for trEPR were collimated into the cryostat and resonator windows from a multi-mode optical fiber, ThorLabs FT600UMT, output and depolarized with an achromatic depolarizer. Light pulses (2 mJ energy, 7 ns duration) at a wavelength of 532 nm were produced by a GWU VersaScan Optical Parametric Oscillator (OPO) pumped by a Newport/Spectra Physics Lab 170 Quanta Ray Nd:YAG pulsed laser operating at 20 Hz, $\lambda = 355$ nm. The trEPR spectra were recorded by direct detection with transient recorder without Lock-In amplification. The trEPR direct-detected signal was recorded through a Bruker SpecJet II transient recorder with timing synchronization by a Stanford Research Systems DG645 delay generator. The overall response time of the instrument was about 200 ns.

From the data set obtained, the transient EPR spectrum at 1 μs after the laser pulse was extracted, which is an appropriate compromise for all samples. The reported trEPR spectra were averaged across a temporal time window of 0.4 μs (0.8–1.2 μs). The acquired trEPR spectra were simulated by using the core functions pepper and esfit of the open-source MATLAB toolbox EasySpin[73]. The parameters included in our best-fit simulations are the absolute values of ZFS parameters ($|D|$ and $|E|$), the triplet population sublevels both for inter-system crossing (ISC) ($p_x$, $p_y$, $p_z$), the relative weight between ISC and back electron transfer (BET) triplets, and the line broadening of both ISC and BET triplets (assumed as only Gaussian to not over-parametrize the fitting). For ISC triplets, the populations of the triplet sublevels at zero field were calculated ($T_x$, $T_y$, $T_z$) in the fitting program and used by EasySpin to simulate the trEPR spectrum at resonant fields. For BET triplets the population sublevels at resonant fields were set and kept fixed: $p_{-1} = 1$, $p_0 = 0$, $p_{+1} = 1$, as discussed below. For all simulations, the g-tensor was assumed isotropic with $g_{iso} = 2.002$. Further details on the simulation program and the spin physics of triplet states in organic solar cells can be found in literature[49,51].

### TAS experimental details

The transient absorption (TA) setup used in this work features a Ti:Saphire laser (Coherent, 800 nm, 1 kHz repetition rate, 100 fs pulse duration). The output is split into the pump and probe beam paths. The probe path generates an idler of 2000 nm in a home-built two-pass OPA which is used to pump a YAG crystal, generating white-light from 650–1700 nm. The pump path utilizes a home-built NOPA to generate an excitation pulse centered at 530 nm. The excitation density was below $2.0 \times 10^{19}$ cm$^{-3}$.

### Reporting summary

Further information on research design is available in the Nature Portfolio Reporting Summary linked to this article.

## Data availability

All source data generated in this study have been deposited on zenodo at https://zenodo.org/records/10042112.

## Code availability

The code used to analyze the data reported in this study are available from the corresponding authors upon request.

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

## Acknowledgements

A.J. acknowledges funding from the Wolfson-Marriott Graduate Scholarship from Wolfson College, Oxford, the EPRSC Doctoral Training Accounts, and the Department of Physics at the University of Oxford. E.C., V.C., S.M.P., and J.L.B. acknowledge funding from UA College of Science and the Office of Naval Research, Award No. N00014-20-1-2110, and the use of High-Performance Computing (HPC) resources supported by Research Data Center (RDC) at the University of Arizona. A.P. acknowledges funding from the European Union's Horizon 2020 research and innovation program under the Marie Skłodowska-Curie grant agreement No. 101104276 (PHOTOCODE). N.B. and K.Y. thank the Swiss National Science Foundation (grant 200020_215384) and the University of Bern for financial support. K.Y. acknowledges funding from the Swiss National Science Foundation (200020\_184819). P.K., T.D., and M.R. acknowledge funding from the Global Challenges Research Fund (GCRF) through Science & Technology Facilities Council (STFC), Grant No. ST/R002754/1: Synchrotron Techniques for African Research and Technology (START). P.K. also thanks EPSRC for funding of a Postdoctoral Fellowship EP/V035770/1. A.E.L. thanks the EPSRC for funding through the Doctoral Training Partnership EP/N509711/1 as well as the STFC, ISIS Neutron and Muon facility and studentship 1948713. I.H. acknowledges funding from the FirstRand Foundation and the Oppenheimer Memorial Trust. GIWAXS measurements were performed at the I07 beamline at the Diamond Light Source (experimental session NT26630-1) and at XMaS: The UK Materials Science Facility (BM28) at the European Synchrotron Radiation Facility (ESRF; experimental session 28-01-1274). The data can be accessed at https://data.esrf.fr/ and via https://doi.org/10.15151/ESRF-DC-772678050. We thank Didier Wermeille (University of Liverpool) and Oier Bikondoa (University of Warwick) for their assistance with remote experimentation and advice. The authors also sincerely thank Dr. Andreas Sperlich for the invaluable and fruitful discussions on EPR and photophysics in preparation of this manuscript. This research was funded in whole, or in part, by the UKRI (Grant numbers see above). For the purpose of Open Access, the author has applied a CC BY public copyright license to any Author Accepted Manuscript version arising from this submission.

## Author contributions

A.J., P.K., and M.R. designed and coordinated the project. A.J. carried out the device characterization, E.C., S.M.P., and V.C. performed the DFT calculations, A.P. the EPR measurements, K.Y. the TAS measurements, A.E.L. and T.D. the GIWAXS measurements, and S.V.K. and I.H. performed the ellipsometry measurements. Work at the respective institutions was overseen by N.B., J.L.B., and M.R. All authors contributed to the analysis and manuscript writing.

## Competing interests

M.R. is founder of TerraChange Solar, a start-up in the space of organic PV. The remaining authors declare no competing interests.
