## [Peer Review File · Nature Communications]

Limiting factors for charge generation in low-offset fullerene-based organic solar cellsREVIEWER COMMENTS

Reviewer #1 (Remarks to the Author):

The manuscript by Jungbulth et al. investigated the effect of the fluorination of ZnPc on the properties of local excitonic (LE) and charge transfer (CT) singlet/triplet states, with an aim to better understand the mechanism behind free charge generation in low-offset organic photovoltaics. The targeted fluorination in ZnPc enables the fine tuning of the energy levels, resulting in a series of donor:C60 blends with varying HOMO-HOMO offsets. The role of HOMO-HOMO offset between donor and acceptor is an active area in OPV field, as many studies have shown that the trade-off between voltage loss and charge generation exists when varying the offset. Understanding the function of the offset is therefore necessary for further improving the OPV performance beyond 20%.

In this work, by studying a series of fluorinated ZnPc blending with C60, the authors discovered that the charge generation is limited by the CT-CS transition rather than LE-CT transition, after considering the impact of back transfer from CT to triplet states through trEPR experiments. This is an interesting observation and highlights the important role of triplet states in understanding the charge generation process, and will certainly stimulate more research efforts into triplet states in OPVs.

The manuscript is sound and well written, I therefore recommend publication with minor revisions:

1) Could the authors quantify the different components of voltage losses for each blend device, i.e. radiative vs nonradiative? This is relevant to the question regarding the trade-off between voltage losses and charge generation (although the results seem obvious).

2) Related to the question above, BET from CT to triplet states reduces the charge generation yield in e.g. F8ZnPC:C60 relative to ZnPc:c60, but should also increase nonradiative recombination losses (following Nature volume 597, pages666–671 (2021)). If I'm not mistaken about the nonradiative voltage loss values (for the authors to check), F8ZnPC:C60 shows lower nonradiative voltage losses than ZnPc:C60, which seems contradictory to Nature volume 597, pages666–671 (2021). Could the authors clarify this point?

3) Could the author comment on why the offset could be the same for all blends, and the effect of CT-CS offset on the charge generation yield? In a recent paper in Energy Environ. Sci., 2022, 15, 1256-1270, Nelson et al. observed that the CT-CS offset changes with LE-CT offset through model fitting. In this manuscript, the offset was set the same for all four blends. I'm not sure how reliable that is, but certainly it would impact the charge generation yield if there was a difference.

Reviewer #2 (Remarks to the Author):

Jungbluth and co-authors report in this manuscript (“Limiting factors for charge generation in low-offset fullerene-based organic solar cells”) the investigation of organic solar cells consisting of donor ZnPc and acceptor fullerene C60. By changing the fluorination level (ZnPc, F4ZnPc, F8ZnPc, F16ZnPc) of the donor material, they change the HOMO and LUMO levels approximately equally, keeping the optical absorption similar, but changing the energetics with respect to the interfacial CT state involving C60. They manufacture organic cells, with a low donor content (wt.5% ZnPc / fluorinated derivatives and wt.95% C60) to prevent recombination, and characterize their devices with several methods including UV-vis spectroscopy, current density-voltage measurements and time-resolved EPR spectroscopy (trEPR). These experimental methods are complemented by theoretical calculations on the DFT level and TDDFT level. They conclude that “CT dissociation rather than CT formation” presents a bottleneck for free charge generation in fullerene containing organic solar cells with low energetic differences between LE and CT states.

Overall, I consider the results interesting and the manuscript largely ready for publication. Quality of experiments and computational work is good, and presentation of the results and the discussion seems in general appropriate for the audience of Nature Communications. However, I have two major points of criticism. First, in my opinion the work needs some time-resolved data to support their (kinetic) models for the four different systems (elaborated in more details below). Second, the manuscript may not fulfill the very high expectations for Nature Communications, and thus could be more suitable for a different, more specialized journal in fields like physical chemistry or materials chemistry. While title and abstract convey the message that some new and important general knowledge about the intricacies of charge separation in fullerene containing OPV active materials is learned, the study itself is essentially completely focused on their one specific type of system (specific donor and acceptor molecules, their ratio, and the morphology of the blend) and it is not clear to me how much of these findings can be generalized.

Specific Issues

While trEPR is a very powerful technique and allows to provide unique insight into the weak magnetic interactions between electron spins and with the external magnetic field, it is lacking in quantitative power (e.g. how many CT and CS states are generated per absorbed photon) and in time-resolution (100s of nanoseconds). The authors heavily base their (kinetic) model on the trEPR data, but kinetic data from trEPR are not shown, not even in the SI. Only a single time-slice is shown. Here, another time-resolved method like ultrafast UV-vis spectroscopy could have been useful and provide very valuable complementary information. The deviation of the BET mechanism in trEPR from the typical S-T0 generation is a key result and should be mentioned in the main manuscript, not only in the SI. However, this needs to be discussed taking into account the knowledge about the sign of ZFS-parameter D, and if D is indeed positive. Certainly the ZFS parameters can be computed for monomeric ZnPc and its fluorinated derivatives to support or contradict this conclusion.

While the authors state that two types of signals in the trEPR spectra are observed, the SCRP signals are not well visible in Figure 4; the triplet states dominate the spectra. Subtraction of the triplet background should be performed to make the SCRP spectra visible. There should be an extra figure showing SCRP spectra either in the manuscript or in the SI. The statement in the figure caption that these SCRP signal are due to photogenerated free charges is misleading. Stable signals from paramagnetic photogenerated charges are subtracted during normal data processing, and the spin-polarized SCRP signals are from centers where charge separation and recombination happens repeatedly upon light excitation.

In the DFT part, it should be discussed how established this methodology is for this type of systems (including references), and what the typical errors are. Calculations are performed only for donor:C60 pairs, and then corrections are applied. Considering the expertise of the computational group I would expect that a few calculations should have been performed for systems with 2 or 3 C60 molecules to obtain an estimate of the magnitude of correction for interactions of the donor ZnPc with more than one fullerene acceptor.

Some minor specific issues with the manuscript are listed below:

- PCE values are important but not provided in the manuscript. The authors state the PCE values are in the SI, but they are not listed but have to be calculated by the reader from the table. These values should be provided by the authors, and they should be mentioned in Figure 1 or in the main text.
- There are several error messages in the manuscript (“Error! Reference source not found”)
- It is reported that measurement were performed on a Bruker E680 X-band, which is a contradiction. The E680 is a W-band spectrometer, and all data presented are X-band data.
- Details about the partial ordering in the EPR should be provided in the SI.
- What is the justification for using Lorentzian instead of Gaussian broadening for modelling the EPR spectra in these systems? Strain effects of the D-tensor and unresolved hyperfine interactions should lead to Gaussian broadening.
- The signs and magnitude of the ZFS-parameters D and E should be clearly distinguished by using $|D|$ and $|E|$ or providing a sign for D and E.
- The discussion concerning D as a measure of delocalization in this non-linear systems is taken a bit far and too simplified, since the different surroundings in the films can also influence D and should be discussed as possible contributor.
- Donor and acceptor spins interact not only due to exchange but also spin-spin dipolar mechanisms.
- Which resonator was used for the EPR measurements? This information is lacking.
- References are a bit sparse in certain places and too focused on previous work from some groups.

Reviewer #3 (Remarks to the Author):

The focus of this manuscript is on providing more fundamental insights on the free charge generation process in organic solar cells. The authors claimed that the CT dissociation is the major bottleneck for free charge generation in fullerene-based blends where the energy offset between local excitation (LE, singlet) and CT state is small.

The authors have done a very nice job in writing up the introduction to set the stage for this work. Using diluted donor in acceptor (5:95 in this case) is not a new idea, but an interesting method to focus on some fundamental process. However, I am not entirely sure the working mechanism would be BHJ at all in this case, since both C Tang's work (the authors cited) and Jinsong Huang's work (Adv. Mater. 2013, 25, 572–577) sort of indicated that the working mechanism was Schottky junction. Please provide some justification on using diluted donor in acceptor, and how the working mechanism would impact (or not) the focus of the study.

My main concern is that, after reading the entire paper a few times, I am still not convinced.

1. Data wise, Figure 1 is the I-V curve, Figure 2 is DFT, Figure 3 is EQE and Figure 4 is the EPR. Figure 5 is the schematic diagram. With the data presented, in particular, missing the TA data, I do not think the authors can draw a proper energy diagram as shown in Figure 5. Much is missing on the non-geminate triplet. One can argue that he/she can focus on geminate triplet, but without the full picture, it is hard to understand the delta CT vs delta CS, and the relative energy levels.
2. If the authors are trying to argue the importance of CT dissociation (which I agree), shouldn't the authors need to measure the $E(\text{CT})$ and $E(\text{CS})$? It seems that all these data are from calculation.
3. Table 2 shows that the delta $E(\text{CS})$ for all four is similar at -0.5 eV. Then the authors said that it was constant as $E(\text{el})$. Delta $E(\text{CS})$, as the authors defined in the introduction, is the difference between $E(\text{CT})$ and $E(\text{CS})$. Now it is a constant?
4. Figure 5 might be a good idea in general; but with all these arrows, I am not entirely sure what the authors meant.
5. In general, higher CT state does not necessarily translate into high Voc; the back electron transfer to the T1 (non emissive state) is a major loss of Voc (via non radiative decay), as the 2021 Nature showed.

Furthermore, the paper is hard to read. Many comparisons are convoluted and I have had a hard time to follow the writing. The 2021 Nature paper (<https://doi.org/10.1038/s41586-021-03840-5>) that the authors cited did a great job in writing. That paper has even MORE systems than this one,

but it is very easy to follow the idea and discussion. I think the authors really need to cut down some text and condense the writing to focus on the delta CT and delta CS.

A minor issue: the authors need to fix many “Error! Reference source not found.”

With all these reasons, I believe publishing this work at this stage is pre-mature. The authors need more data and better writing to provide a compelling story.

Reviewer #1 (Remarks to the Author):

The manuscript by Jungbluth et al. investigated the effect of the fluorination of ZnPc on the properties of local excitonic (LE) and charge transfer (CT) singlet/triplet states, with an aim to better understand the mechanism behind free charge generation in low-offset organic photovoltaics. The targeted fluorination in ZnPc enables the fine tuning of the energy levels, resulting in a series of donor:C60 blends with varying HOMO-HOMO offsets. The role of HOMO-HOMO offset between donor and acceptor is an active area in OPV field, as many studies have shown that the trade-off between voltage loss and charge generation exists when varying the offset. Understanding the function of the offset is therefore necessary for further improving the OPV performance beyond 20%. In this work, by studying a series of fluorinated ZnPc blending with C60, the authors discovered that the charge generation is limited by the CT-CS transition rather than LE-CT transition, after considering the impact of back transfer from CT to triplet states through trEPR experiments. This is an interesting observation and highlights the important role of triplet states in understanding the charge generation process, and will certainly stimulate more research efforts into triplet states in OPVs.

The manuscript is sound and well written, I therefore recommend publication with minor revisions:

1) Could the authors quantify the different components of voltage losses for each blend device, i.e. radiative vs. nonradiative? This is relevant to the question regarding the trade-off between voltage losses and charge generation (although the results seem obvious).

Response: We agree that a discussion of voltage losses was needed in the manuscript. We have now added additional information and figures in the Supplementary Information. More specifically, we added **section B.2**, which discusses how we calculate voltage/energy losses, **Figure S10** which demonstrates the importance of measuring the EQE with high dynamic range to accurately determine the radiative upper limit of the open circuit voltage, **Table S3** which documents the calculated losses, **Figure S11** which plots the calculated energy levels and losses, and finally **Figure S12** which shows how radiative and non-radiative losses relate to the measured CT state energies.

2) Related to the question above, BET from CT to triplet states reduces the charge generation yield in e.g., F8ZnPc:C60 relative to ZnPc:C60, but should also increase nonradiative recombination losses (following Nature volume 597, pages 666–671 (2021)). If I'm not mistaken about the nonradiative voltage loss values (for the authors to check), F8ZnPc:C60 shows lower nonradiative voltage losses than ZnPc:C60, which seems contradictory to Nature volume 597, pages 666–671 (2021). Could the authors clarify this point?

Response: Our trEPR and TAS measurements allow us to study geminate and non-geminate triplet formation in thin films (where all charges recombine eventually). With this, we can probe different exciton decay pathways which provide deconvoluted information on the $S_1 - CT$ and $CT - CS$ transitions which are essential for free charge generation. However, in addition to recombination after triplet formation, the energy gap law is also at play, which states that non-radiative coupling to the ground state decreases with increasing E_{CT} as more vibrational quanta would be required. Looking at the trends of nonradiative energy losses vs. E_{CT} of the devices (**Figure S12a**), we observe lower non-radiative losses for $F_8ZnPc:C_{60}$ than $ZnPc:C_{60}$ as correctly identified by the reviewer, and as predicted by the energy gap law. In line with the framework presented in Nature volume 597, pages 666–671 (2021), specifically equation 2 therein, the triplet formation pathways are captured by χ , i.e., the fraction of radiative recombination events), while a higher E_{CT} influences Φ_{PL} (i.e., the photoluminescence quantum efficiency) via the energy gap law. In the end, we mainly focus on the pathways for free charge generation as this is the main barrier in understanding low-offset systems. To improve this discussion in the paper, we added an additional paragraph to the voltage loss section in the Supplementary Information.

3) Could the author comment on why the offset could be the same for all blends, and the effect of CT-CS offset on the charge generation yield? In a recent paper in Energy Environ. Sci., 2022, 15, 1256–1270, Nelson et al. observed that the CT-CS offset changes with LE-CT offset through model fitting. In this manuscript, the offset was set the same for all four blends. I'm not sure how reliable that is, but certainly it would impact the charge generation yield if there was a difference.

Response: We thank the reviewer for their question. The main aim of the calculations leading to the results referred to by the reviewer and shown in former **Table 2** (now **Table 1**) was to provide an alternative estimate of the CT energies. We used a simple model where the CS and CT states differ by the hole-electron electrostatic interaction energy, which we assumed to be the same for all 4 investigated blends (this is a reasonable assumption since the donor content is small, and all systems should have approximately the same dielectric constant). Of course, when one compares systems with different electronic polarization energies and different dielectric constants, a change in, for instance, the donor IP will change both the CT-CS and LE-CT offsets.

To clarify this point in the main text, we added the following paragraph:

“In donor-acceptor blends, E_{el} depends on the dielectric constant and the interatomic distance of localized electrons/holes and is expected to fall between -0.3 eV and -0.6 eV. For the calculations presented in this work, we assume that E_{el} is equal to -0.50 eV and constant for all four blends. [1]–[3]

This is a reasonable assumption given the low donor content of each blend that should result in comparable dielectric constants for all four material system.”

Reviewer #2 (Remarks to the Author):

Jungbluth and co-authors report in this manuscript (“Limiting factors for charge generation in low-offset fullerene-based organic solar cells”) the investigation of organic solar cells consisting of donor ZnPc and acceptor fullerene C60. By changing the fluorination level (ZnPc, F4ZnPc, F8ZnPc, F16ZnPc) of the donor material, they change the HOMO and LUMO levels approximately equally, keeping the optical absorption similar, but changing the energetics with respect to the interfacial CT state involving C60. They manufacture organic cells, with a low donor content (wt.5% ZnPc / fluorinated derivatives and wt.95% C60) to prevent recombination and characterize their devices with several methods including UV-vis spectroscopy, current density-voltage measurements and time-resolved EPR spectroscopy (trEPR). These experimental methods are complemented by theoretical calculations on the DFT level and TDDFT level. They conclude that “CT dissociation rather than CT formation” presents a bottleneck for free charge generation in fullerene containing organic solar cells with low energetic differences between LE and CT states.

Overall, I consider the results interesting and the manuscript largely ready for publication. Quality of experiments and computational work is good, and presentation of the results and the discussion seems in general appropriate for the audience of Nature Communications. However, I have two major points of criticism. First, in my opinion the work needs some time-resolved data to support their (kinetic) models for the four difference systems (elaborated in more details below).

Response: We thank the reviewer for their positive evaluation of our manuscript, and the constructive feedback regarding adding time-resolved measurements. We completely agree that adding time-resolved data strengthens the manuscript. As discussed in more detail below, we have now added both time-resolved EPR, and transient absorption measurements which fully support and strengthen our interpretation of the kinetics in our studied blends.

Second, the manuscript may not fulfill the very high expectations for Nature Communications, and thus could be more suitable for a different, more specialized journal in fields like physical chemistry or materials chemistry. While title and abstract convey the message that some new and important general knowledge about the intricacies of charge separation in fullerene containing OPV active materials is learned, the study itself is essentially completely focused on their one specific type of system (specific

donor and acceptor molecules, their ratio, and the morphology of the blend) and it is not clear to me how much these findings can be generalized.

Response: We thank the reviewer for drawing our attention to the fact that the generality of our experiments could have been stressed better. We believe that our selection of samples is not a limitation. Firstly, the studied material systems allow a systematic study of how shifting molecular energy levels influence charge generation and recombination. By selecting this donor fluorination series, we were able to minimize differences in molecular properties (like absorption) between the different samples. By making samples with low donor content, we were also able to exclude the influence of changing microstructure, which is often overlooked and can have a significant impact on optoelectronic properties. In the context of OPV research more generally, significant knowledge has been derived from dilute systems, and was later found to apply to all material systems and D:A mixing ratios (e.g. the seminal work of Benduhn *et al.* on the energy gap law for OPV ‘Intrinsic non-radiative voltage losses in fullerene-based organic solar cells’. *Nat Energy* **2**, 17053 (2017)). While dilute systems have reduced complexity that allow systematic studies, their interface energetics are representative of BHJs, especially for charge generation and recombination as studied in our manuscript.

Furthermore, we believe that the conclusions of our work are timely and highly relevant to the OPV community and readers of Nature Communications. Firstly, we highlight the importance of considering the CS state to obtain a full picture of the charge dynamics in OPV material systems. For too long, the CS state has been left out of the discussion, and most research only focused on understanding and optimizing CT states and S₁-CT transitions. Our work challenges this standard picture of free charge generation and highlights that CT-CS transitions rather than the S₁-CT transitions can present a bottleneck for free charge generation. This result echoes recent work published for non-fullerene acceptors that employs a very different methodology (e.g., Azzouzi *et al.* ‘Reconciling models of interfacial state kinetics and device performance in organic solar cells: impact of the energy offsets on the power conversion efficiency.’ *Energy & Environmental Science*, **15**, (2022), or Müller *et al.* ‘Charge-Transfer State Dissociation Efficiency Can Limit Free Charge Generation in Low-Offset Organic Solar Cells’, *ACS Energy Letters* **2023** *8* (8), 3387-3397) and presents, in our opinion, a new and important understanding about the charge generation process in OPV.

In terms of generality and comparability to other studies, we would also like to highlight that the breadth of calculations and measurement techniques presented in this manuscript, which we made even more comprehensive in this revision (specifically the combination of DFT, trEPR, and TAS) matches the complexity of recently published work in other high impact journals, e.g., Gillett *et al.* The role of charge recombination to triplet excitons in organic solar cells. *Nature*, volume 597, pages 666–671 (2021). This allows drawing interesting parallels, for instance considering the presence of geminate recombination in our fullerene-based blends, compared to the absence of geminate recombination in the NFAs studied by Gillett *et al.*, despite similar non-radiative losses across both studies. Especially

considering our recent addition of TAS measurements to the manuscript, we are convinced that our work will stimulate more research into the influence of triplets and CS on device performance, which will be highly relevant for advancing the field.

Specific Issues

While trEPR is a very powerful technique and allows to provide unique insight into the weak magnetic interactions between electron spins and with the external magnetic field, it is lacking in quantitative power (e.g. how many CT and CS states are generated per absorbed photon) and in time-resolution (100s of nanoseconds). The authors heavily base their (kinetic) model on the trEPR data, but kinetic data from trEPR are not shown, not even in the SI. Only a single time-slice is shown. Here, another time-resolved method like ultrafast UV-vis spectroscopy could have been useful and provide very valuable complementary information.

Response: We thank the reviewer for their constructive feedback and the good suggestion of providing further time-resolved analysis and methods to support and strengthen our conclusions. We address their comment by including both time-resolved analysis of the already acquired trEPR spectra and by performing new transient absorption measurements to complement and extend our trEPR analysis. Our new results fully support our previous findings.

As for the trEPR analysis, we have added 2D trEPR spectra (**Figure S17**), which we also report below for convenience, and 1D trEPR spectra at three different delays after the laser flash (**Figure S18**). Further analysis and details on the time evolution of the trEPR spectra is reported in the Supplementary Information.

Figure S1. Two-dimensional time-resolved electron paramagnetic resonance (trEPR) spectra of all neat (first column) and dilute donor (second column) films studied in this work. The trEPR spectra were acquired at 80 K and after excitation at 532 nm. In this figure, emission is highlighted in blue, and absorption is highlighted in red.

Furthermore, we performed transient absorption measurements of the dilute blends – included as **Figure 4** in the main paper and **Figure S22** in the SI, and below for convenience. By comparing the spectra of the blends to measurements of neat films (**Figure S23** in the SI) we were able to disentangle the different spectral contributions and further study the charge dynamics in our blends. A detailed discussion of our TAS results is included in the main paper.

Figure 1. Transient absorption (TA) spectra and charge dynamics. The TA spectra were measured at selected time delays after excitation at 530 nm for a) ZnPc:C₆₀ and b) F₈ZnPc:C₆₀ dilute donor blends. TA measurements of F₄ZnPc:C₆₀ and F₁₆ZnPc:C₆₀ are included in **Supplementary Figure S22**. The inset shows the early spectrum of a neat C₆₀ film for reference. c) Dynamics at 1600 nm (mainly C₆₀ excited state absorption (ESA), smooth lines) and at 690 – 720 nm (maximum of the donor ground state bleaching (GSB), dotted lines) for the four investigated F_xZnPc:C₆₀ blends. d) The same dynamics are scaled between 0 and 1 for better comparison. Thick smooth solid lines are multi-exponential fits to the dynamics at 1600 nm, while thick dotted lines are fits to the dynamics at 690 – 720 nm.

Figure S2. **Transient absorption (TA) spectra of fluorinated blends.** The measurements were performed for 0.2 – 1500 ps after excitation for dilute a) $F_4ZnPc:C_{60}$, and b) $F_{16}ZnPc:C_{60}$. All samples show an increase in the ground-state bleach (GSB) of the donor between 0.2 and 50 ps. As the initial excitation at 530 nm mostly excites C_{60} (see **Error! Reference source not found.**), this increase in GSB of the donor is indicative of electron and/or energy transfer from the excited C_{60} to the donor.

The deviation of the BET mechanism in trEPR from the typical S-T0 generation is a key result and should be mentioned in the main manuscript, not only the SI. However, this needs to be discussed taking into account the knowledge about the sign of ZFS-parameter D, and if D is indeed positive. Certainly, the ZFS parameters can be computed for monomeric ZnPc and its fluorinated derivatives to support or contradict this conclusion.

Response: We have now added **Table S10** in the Supplementary Information, which summarizes the DFT derived zero-field splitting parameters. From our DFT calculations, we determined the sign of the D and E values to be positive for all donor molecules. This confirms the [1 0 1] polarization pattern of the BET triplets, which was previously observed and discussed in literature, e.g., Righetto *et al.* ‘Engineering interactions in QDs–PCBM blends: a surface chemistry approach’, *Nanoscale*, 10, (2018), and Franco *et al.* ‘Time-resolved EPR of Photoinduced Excited States in a Semiconducting Polymer/PCBM Blend’, *J. Phys. Chem. C*, (2012).

Considering the feedback from other reviewers, who recommended to limit the technical content of the manuscript, we chose to mention this result only briefly in the main text. A detailed discussion can be found in the Supplementary Information.

While the authors state that two types of signals in the trEPR spectra are observed, the SCRCP signals are not well visible in Figure 4; the triplet states dominate the spectra. Subtraction of the triplet background should be performed to make the SCRCP spectra visible. There should be an extra figure showing SCRCP spectra either in the manuscript or in the SI. The statement in the figure caption that these SCRCP signal are due to photogenerated free charges is misleading. Stable signals from paramagnetic photogenerated charges are subtracted during normal data processing, and the spin-polarized SCRCP signals are from centers where charge separation and recombination happens repeatedly upon light excitation.

Response: We thank the reviewer for pointing out that the statements in the figure caption and in the text were misleading. As the reviewer correctly noted, long-lived charges, e.g., “trapped” polarons, are subtracted during data processing. The narrow EPR signal that we see in the middle of the spectrum can be attributed to either photogenerated separated charges that quickly recombine or to radical pairs ($r < 30 \text{ \AA}$) where the two unpaired spins interact via both dipolar and exchange coupling. Usually, in the latter case, primary radical pairs are not directly observable by EPR due to their short lifetime ($\leq \text{ns}$), and only secondary, well-separated and longer-lived radical pairs can be observed [Adv. Energy Mater. 2017, 7, 1602226]. Since the signal-to-noise ratio of the narrow signal is quite low, it is difficult to easily draw conclusions from our spectra. However, considering that in the blends and in some neat films we observe geminate recombination triplets, we concur that most probably the narrow signal can be attributed to interacting radical pairs ($r < 30 \text{ \AA}$).

As the reviewer suggested we have added a new figure in the SI (**Figure S18**) where we magnified the trEPR spectra of the radical pair region at three different delays from the laser pulse, namely 1 μs (blue line), 2.2 μs (red line), 3.4 μs (orange line). The triplet background has been subtracted via a first-order baseline to make the radical pair spectra more visible. For convenience, we report the new figure and the caption in the following:

Figure S3. **One-dimensional time-resolved electron paramagnetic resonance (trEPR) spectra of all neat (first and second column) and dilute donor (third and fourth column) films.** The trEPR spectra were acquired at 80 K and after excitation at 532 nm. Absorption (*a*) is up, and emission (*e*) is down. For each sample, three representative time delays after the initial laser pulse are reported. Legend: blue = 1 μ s, red = 2.2 μ s, orange = 3.4 μ s. For each sample, the left figure (first and third column) shows the full magnetic field sweep to highlight the time evolution of the triplet states, while the right figure (second and fourth column) shows a narrow sweep magnification of the radical pair signal.

In our spectra, the typical EAEA or AEAE pattern of spin-correlated radical pairs is not observable. This could be due to several reasons, e.g. (1) rapid spin-relaxation [J. Phys. Chem. Lett. 2014, 5, 1, 30–35], (2) distortions due to a sequential electron transfer process [J. Phys. Chem. B 2015, 119, 7407–7416], (3) alternative spin polarization mechanisms [J. Phys. Chem. C 2013, 117, 4, 1554–1560], or more trivially (4) weak signal-to-noise ratio.

We believe that a detailed discussion of the spin-polarization of the radical pairs is beyond the scope of our article, also considering the low signal-to-noise ratio of the radical pair region, and instead decided to focus our attention primarily on the triplet states. Nevertheless, we have included a short discussion of the radical pairs in the Supporting Information for the interested reader.

In the DFT part, it should be discussed how established this methodology is for this type of systems (including references) and what the typical errors are. Calculations are performed only for donor:C₆₀ pairs, and then corrections are applied. Considering the expertise of the computational group I would expect that a few calculations should have been performed for systems with 2 or 3 C₆₀ molecules to obtain an estimate of the magnitude of correction for interactions of the donor ZnPc with more than one fullerene acceptor.

Response: We agree with the reviewer that the investigation of electron delocalization would provide additional insight. However, it should be expected that this effect is similar across all investigated systems. It is reasonable to assume that the main factor that affects the performance of a F_xZnPc:C₆₀ blend in the devices is the position of the CT state and its interaction with LE states. Therefore, we chose to focus on computing the electronic couplings among various states and on vibronic simulations of the CT bands. Going beyond donor:C₆₀ pairs in such calculations and simulations is very challenging and goes beyond the scope of our paper.

Some minor specific issues with the manuscript are listed below:

- PCE values are important but not provided in the manuscript. The authors state the PCE values are in the SI, but they are not listed by have to be calculated by the reader from the table. These values should be provided by the authors, and they should be mentioned in Figure 1 or in the main text.

Response: We have now added the PCE values in **Table S1** in the Supplementary Information, and added the following sentence in the main text:

*“An overview of the performance trends (V_{oc} , J_{sc} , FF , and PCE , including sample statistics) is shown in **Supplementary Table S1** and **Supplementary Figure S3**.“*

- There are several error messages in the manuscript (“Error! Reference source not found”)

Response: We apologize for any broken references. We went through the paper and Supplementary Information again and will ensure all errors are fixed at submission.

- It is reported that measurement were performed on a Bruker E680 X-band, which is a contradiction. The E680 is a W-band spectrometer, and all data presented are X-band data.

Response: We thank the reviewer for drawing our attention to this inconsistency. The measurements were performed on a Bruker E580 spectrometer. We have updated the manuscript with the correct information.

- Details about the partial ordering in the EPR should be provided in the SI.

Response: Following the reviewer's suggestion, we have now added **section D.3** "Partial Preferential Order" of Triplet States in the Supporting Information.

- What is the justification for using Lorentzian instead of Gaussian broadening for modelling the EPR spectra in these systems? Strain effects of the D-tensor and unresolved hyperfine interactions should lead to Gaussian broadening.

Response: The reviewer is right, apologies for missing this. Since the measurements are performed on evaporated films at low temperatures, EPR lines are inhomogeneously broadened due to the presence of several unresolved contributions from anisotropic magnetic interactions. In fact, for our EPR simulations, the Easypin command "Sys.lw" was used, which refers to the FWHM of Gaussian broadening expressed in mT. We amended the text highlighting that only Gaussian broadening has been used for EPR simulations, which is – as the reviewer noted – the appropriate approach for describing the broadening in our systems.

- The signs and magnitude of the ZFS-parameters D and E should be clearly distinguished by using |D| and |E| or providing a sign for D and E.

Response: As briefly mentioned above, we performed DFT calculations at various levels of theory and obtained the following D and E values:

Table S101. **Density functional theory (DFT) derived zero-field splitting parameters.**

	B3LYP/6-31G**		PBE0/6-31G**		B3LYP/def2-TZVP		B3LYP/EPR-II def2-TZVP for Zn	
	D [cm ⁻¹]	E/D	D [cm ⁻¹]	E/D	D [cm ⁻¹]	E/D	D [cm ⁻¹]	E/D
ZnPc	2.676	0.003	2.674	0.005	0.961	0.016	0.379	0.022
F ₄ ZnPc	2.618	0.003	2.582	0.003	0.985	0.014	0.380	0.020
F ₈ ZnPc	2.618	0.002	2.580	0.003	0.797	0.024	0.385	0.027
F ₁₆ ZnPc	2.597	0.002	2.553	0.003	0.765	0.024	0.406	0.006

The calculated values, which are included as **Table S10** in the Supplementary Information, allow us to confirm that the D and E parameters are positive for all studied molecules at all levels of theory.

- The discussion concerning D as a measure of delocalization in this non-linear systems is taken a bit far and too simplified, since the different surroundings in the films can also influence D and should be discussed as possible contributor.

Response: Many thanks for highlighting this. As the reviewer correctly says, the dielectric environment can play a pivotal role on the triplet wavefunction and in turn on its ZFS values. Furthermore, we agree that it is difficult to draw any conclusions about triplet delocalization when comparing different F_xZnPc molecules, which have intrinsically different chemical structures and thus triplet wavefunctions. To this end, we therefore corrected the discussions in the Supporting Information and also updated it in view of the results of the theoretical calculations.

- Donor and acceptor spins interact not only due to exchange but also spin-spin dipolar mechanisms.

Response: Thank you for pointing out this mistake. We have updated the discussion on SCRIP in the Supporting Information following the reviewer comment.

- Which resonator was used for the EPR measurements? This information is lacking.

Response: We thank the reviewer for bringing this omission to our attention. The resonator that was used for the trEPR measurements is the X-band Dielectric Resonator (ER 4118X-MD5). This information has been added to the text.

References are a bit sparse in certain places and too focused on previous work from some groups.

Response: We thank the reviewer for their feedback. Throughout the editing process, we added additional references to the manuscript, e.g. Müller *et al.* ‘Charge-Transfer State Dissociation Efficiency Can Limit Free Charge Generation in Low-Offset Organic Solar Cells’, *ACS Energy Letters* 2023 8 (8), 3387-3397, which is highly relevant and was published recently.

Reviewer #3 (Remarks to the Author):

The focus of this manuscript is on providing more fundamental insights on the free charge generation process in organic solar cells. The authors claimed that the CT dissociation is the major bottleneck for free charge generation in fullerene-based blends where the energy offset between local excitation (LE, singlet) and CT state is small.

The authors have done a very nice job in writing up the introduction to set the stage for this work. Using diluted donor in acceptor (5:95 in this case) is not a new idea, but an interesting method to focus on some fundamental process. However, I am not entirely sure the working mechanism would be BHJ at all in this case, since both C Tang’s work (the authors cited) and Jinsong Huang’s work (*Adv. Mater.* 2013, 25, 572-577) sort of indicate that the working mechanism was Schottky junction. Please provide some justification on using diluted donor in acceptor, and how the working mechanism would impact (or not) the focus of the study.

Response: We thank the reviewer for their positive feedback on our introduction and methodology. The device physics of dilute OPV remains an interesting field of study and many open questions remain when it comes to understanding their working mechanism, especially their charge transport.

Our work focusses on studying charge generation and recombination, which is governed by the interface, and more specifically by the electronic states of and between neighboring donor and acceptor molecules. In this respect, dilute organic solar cells are representative of BHJs, and their simplified interface microstructure previously allowed the development of breakthrough models such as the energy gap law (Benduhn *et al.* Intrinsic non-radiative voltage losses in fullerene-based organic solar cells. *Nat Energy* 2, 17053 (2017)), which applies to all D:A mixing ratios.

Regarding the precise terminology and calling such devices ‘dilute BHJs’, there most certainly is room for debate. The band diagram, however, is largely unaffected by whether the active layer consists of BHJ or dilute donor blends. A Schottky junction in the sense of a metal/doped semiconductor junction as proposed in the 2013 publication by Yang *et al.* seems unlikely since C₆₀ is not expected to be doped much. If C₆₀ was doped by default, BHJs would also have Schottky-diode band diagrams.

Zhang *et al.*'s 2011 publication discusses a Schottky barrier at the interface between the organic active layer and MoO_x. This is often assumed to be present in OPV and ensures good contact and diode characteristics so that the V_{oc} is governed by the active layer material (as for our studied material systems) and not the contacts.

My main concern is that, after reading the entire paper a few times, I am still not convinced.

1. Data wise, Figure 1 is the I-V curve, Figure 2 is DFT, Figure 3 is EQE and Figure 4 is the EPR. Figure 5 is the schematic diagram. With the data presented, in particular missing the TA data, I do not think the authors can draw a proper energy diagram as shown in Figure 5. Much is missing on the non-geminate triplet. One can argue that he/she can focus on geminate triplet, but without the full picture, it is hard to understand the delta CT vs delta CS, and the relative energy levels.

Response: We thank the reviewer for their feedback. We fully agree transient absorption measurements, including an investigation of non-geminate triplets, should have been added from the start. As detailed in response to reviewer 2, we have now undergone an extensive measurement campaign and added transient absorption measurements of the dilute blends (**Figure 4** and **Figure S22**) and neat films (**Figure S23**). A detailed discussion of our TAS measurements is included in the "Charge Separation and Recombination" section in the main paper. Based on the additional insights gained from TAS, we refined our original energy level diagrams (**Figure 5** and **Figure S24**) and were able to strengthen our original conclusions.

2. If the authors are trying to argue the importance of CT dissociation (which I agree), shouldn't the authors need to measure $E(CT)$ and $E(CS)$? It seems that all these data are from calculations.

Response: Beyond DFT, we measured E_{CT} directly using sensitive EQE measurements. Importantly, we performed EQE fitting using both standard Marcus theory (**Figure S9**, **Table S2**), and the more complex three-state vibronic model (**Figure 2**, **Figure S16**, and **Table S12**). Regarding E_{CS} , to the best of our knowledge, no clear way of measuring E_{CS} is established in literature. Even if direct characterization of E_{CS} was possible, previous work (e.g. Azzouzi *et al.* Nonradiative Energy Losses in Bulk-Heterojunction Organic Photovoltaics, Phys. Rev. X 8, 031055, (2018)) highlights that not only the energies of the CT/CS state are important, but that other factors, like the reorganization energy or entropy, also strongly influence charge transfer. We therefore modified a the relevant sentences in the abstract and introduction to better reflect this complexity.

3. Table 2 shows that the delta E(CS) for all four is similar at -0.5 eV. Then the authors said that it was constant as E(el). Delta E(CS), as the authors defined in the introduction, is the difference between E(CT) and E(CS). Now it is a constant?

Response: We thank the reviewer for their comment. Indeed, in the picture where the energies of the CS and CT states differ by the hole-electron electrostatic interaction energy (E_{el}), the difference between E_{CT} and E_{CS} should be the same if E_{el} is the same. Since the average distances between F_xZnPc and C_{60} are equal for all considered cases and the dielectric constants are the same in all blends (due to the low donor content), having constant $\Delta E_{CS} = E_{CT} - E_{CS}$ values in the presented material systems is reasonable. We note that this does not mean that the charge separation efficiency should also be constant as the transition from CT to CS competes with geminate recombination.

As noted above in response to a similar comment by reviewer 2, we added the following paragraph to the manuscript:

“In donor-acceptor blends, E_{el} depends on the dielectric constant and the interatomic distance of localized electrons/holes and is expected to fall between -0.3 eV and -0.6 eV. For the calculations presented in this work, we assume that E_{el} is equal to -0.50 eV and constant for all four blends. [1]–[3] This is a reasonable assumption given the low donor content of each blend that should result in comparable dielectric constants for all four material system.”

4. Figure 5 might be a good idea in general; but with all these arrows, I am not entirely sure what the authors meant.

Response: We agree with the reviewer that **Figure 5** could have been easier to follow. We therefore decided to simplify all transition diagrams, reduced the number of arrows, and separated the figure into two components, with **Figure 5** in the main paper focusing on $ZnPc$ and F_8ZnPc , and **Figure S26** in the Supplementary Information focusing on F_4ZnPc and $F_{16}ZnPc$.

5. In general, higher CT state does not necessarily translate into high Voc; the back electron transfer T1 (non emissive state) is a major loss of Voc (via non radiative decays), as the 2021 Nature showed.

Response: We fully agree with the reviewer’s comment. To make this point clearer in the manuscript, we have added an extensive characterization of energy losses in the Supplementary Information. More specifically, we added **section B.2**, which discusses how we calculate energy losses, **Figure S10** which demonstrates the importance of measuring the EQE with high dynamic range to accurately determine the radiative upper limit of the open circuit voltage, **Table S3** which documents the calculated energy

losses, **Figure S11** which plots the calculated energy levels and energy losses, and finally **Figure S12** which shows how radiative and non-radiative energy losses relate to the measured CT state energies. As mentioned above in response to a comment by reviewer 1, our trEPR and TAS measurements allow us to study geminate and non-geminate triplet formation in thin films (where all charges recombine eventually). With this, we can probe different exciton decay pathways which provide deconvoluted information on the $S_1 - CT$ and $CT - CS$ transitions which are essential for free charge generation. However, in addition to recombination after triplet formation, the energy gap law is also at play, which states that non-radiative coupling to the ground state decreases with increasing E_{CT} as more vibrational quanta would be required. Looking at the trends of nonradiative energy losses vs. E_{CT} of the devices (**Figure S12a**), we observe lower non-radiative losses with increasing E_{CT} (and donor fluorination), as predicted by the energy gap law. In line with the framework presented in Nature volume 597, pages 666–671 (2021), specifically equation 2 therein, the triplet formation pathways are captured by χ (i.e., the fraction of radiative recombination events), while a higher E_{CT} influences Φ_{PL} (i.e., the photoluminescence quantum efficiency) via the energy gap law. In the end, we mainly focus on the pathways for free charge generation in this manuscript, as this is the main barrier in understanding low-offset systems. To improve this discussion in the paper, we added an additional paragraph to the voltage loss section in the Supplementary Information.

Furthermore, the paper is hard to read. Many comparisons are convoluted and I have had a hard time to follow the writing. The 2021 Nature paper (<https://doi.org/10.1038/s41586-021-03840-5>) that the authors cited did a great job in writing. That paper has even MORE systems than this one, but it is very easy to follow the idea and discussion. I think the authors really need to cut down some text and condense the writing to focus on the delta CT and delta CS.

Response: We thank the reviewer for their feedback and agree that the paper was not as easy to follow as intended and could have been more streamlined. We have taken this comment very seriously and have made a series of modifications to improve the clarity of our manuscript. All changes are tracked in the paper, and are listed below:

Figure Modifications:

- Moved DFT calculations of the HOMO/LUMOs of all molecules from **Figure 2** to **Figure 1** for ease of explanation.
- Removed DFT calculations of molecular and blend energy levels from former **Figure 2** to simplify the narrative of the paper. All values are still reported in **Tables S4 – S7** in the Supplementary Information. With this, former **Figure 2** was removed as a separate figure.

- To reduce repetition, we deleted former **Table 2** in the paper, since the DFT calculations of the energy levels are already included in **Figure 2c** and **Tables S4 – S7**.
- Moved three-state vibronic model fit of $F_4ZnPc:C_{60}$ into Supplementary Information, labelled as **Figure S16**. The main paper now only shows the spectrum and three-state fit for $ZnPc:C_{60}$ as a representative example.
- Moved EPR spectra and fits of $F_4ZnPc:C_{60}$ and $F_{16}ZnPc:C_{60}$ into Supplementary Information, now labelled as **Figure S20**. The main paper now only shows the spectra and fits for $ZnPc:C_{60}$ and $F_8ZnPc:C_{60}$ as representative examples.
- For increased clarity of the manuscript, we moved the transition diagrams for the F_4ZnPc and $F_{16}ZnPc$ blends into the Supplementary Information (now labelled as **Figure S26**) and decided to only show the transition diagrams of the $ZnPc$ and F_8ZnPc blends in the main paper (**Figure 5**).

Text Modifications:

- We made small modifications to the abstract to streamline the narrative and link our work better to the prevalent research questions of the field.
- In response to one of the other reviewer comments, we edited the introduction to clarify that energetic differences between LE, CT, and CS states are only some of the factors influencing transition rates. In addition to energetic offsets, e.g., reorganization energies or competition with GS recombination also influence the efficiency of energetic transitions.
- We edited the device performance section for clarity and moved the technical discussion of the different contributions to the photocurrent to the SI (now section B.1).
- To improve the connection between sections, we restructured the Energy Level discussion and moved the experimental fitting of CT states before the DFT calculations. We moved several technical details into the Supplementary Information and modified **Table 1** to include both the DFT results, and the experimentally fitted CT state energies. We believe that this change improves the clarity of the entire section, as it makes it easier for the reader to follow our narrative and compare the ECT values determined using the different computational and experimental approaches.
- We simplified the section on charge recombination and separation, editing the text for clarity.
- We added an extensive discussion of our newly added transient absorption measurements.
- We moved the discussion of the energy level diagram from the “Charge Separation and Recombination” section to the discussion to separate the presentation of the experimental results and our combined analysis of all findings. Furthermore, we substantially edited and re-wrote the discussion and conclusions, to make our narrative easier to follow and clearly highlight the main take-aways of our manuscript.

A minor issue: the authors need to fix many “Error! Reference source not found.”

Response: We apologize for any broken references. We went through the paper and Supplementary Information again, and will ensure that all errors are fixed upon submission.

With all these reasons, I believe publishing this work at this stage is pre-mature. The authors need more data and better writing to provide a compelling story.

Response: We thank the reviewers for their helpful and constructive feedback. With the extra measurements, along with the substantial changes we made to the structuring and presentation of our results, we believe that our manuscript is now significantly improved, and provides a compelling story to the diverse readership of Nature Communications.

Further changes made to the manuscript:

- Added figure showing alternative EPR spectral fit for $F_{16}ZnPc:C_{60}$, considering only ISC, as **Figure S21** in Supplementary Information.

REVIEWER COMMENTS

Reviewer #1 (Remarks to the Author):

The authors have made tremendous efforts to improve the manuscript, and I agree with the main conclusions, however, there are two points for the authors to consider to enhance the manuscript and to merit a publication in Nature Communications:

1) If a significant amount of CT excitons is lost through triplet states (either through ISC or BET), the voltage loss should be increased, but it does the opposite, as in F8ZnPc:C60 compared to ZnPc:C60. This bothers me a lot. Could the authors clarify?

2) As also noted by Ref. 3, I still find the constant delta ECS a bit unsatisfactory, and the values are negative and very high (-0.5 eV). I'm not sure how efficient the CT dissociation can be for such a negative and high energy barrier as in ZnPc:C60. Also, are the authors implying the same rate constant of CS dissociation (not the yield) for all blends since delta ECS is the same? This would need further justification as well. My feeling is that if the authors want to make a strong statement about CS dissociation, the Delta ECS should be properly investigated as it directly affects the dissociation yield. Clarifying this point could make the conclusions more convincing.

Reviewer #2 (Remarks to the Author):

I really appreciate the effort the authors have made to address the concerns of the reviewers. The additional data provided and their analysis adds substantially in supporting their conclusions, and they clearly improved their manuscript in the revised version. However, I think my second major criticism was only partially addressed. This point was related to how much the findings could be transferred to BHJ systems in general. This is of course a very difficult task to achieve. Overall, I would recommend this manuscript now for publication in Nature Communications.

Reviewer #3 (Remarks to the Author):

The authors have done a very good job in revising the manuscript. I am happy with this version.

We thank the reviewers for their positive feedback and answer the remaining points below:

The authors have made tremendous efforts to improve the manuscript, and I agree with the main conclusions, however, there are two points for the authors to consider to enhance the manuscript and to merit a publication in Nature Communications: 1) If a significant amount of CT excitons is lost through triplet states (either through ISC or BET), the voltage loss should be increased, but it does the opposite, as in F8ZnPc:C60 compared to ZnPc:C60. This bothers me a lot. Could the authors clarify?

Response: We thank the reviewer for their positive evaluation of our manuscript and the changes we implemented. Overall, we believe our data carries greater significance for the photocurrent generation process than the voltage losses, since some understanding of the underlying mechanisms of voltage losses has already been achieved in literature [1], [2].

However, to adequately address the reviewer's comment, we note that recombination via triplet states is not the only pathway for voltage losses and very likely not the dominant one. For ZnPc and F₄ZnPc, the energetic difference between the singlet and CT state causes significant voltage losses (we found 140 – 300 mV). This loss is smaller for F₈ZnPc and F₁₆ZnPc because of the lower energetic offset (shown in section B.2 in the Supplementary Information) and thus reduces the total voltage losses of our material systems.

Indeed, Gillet et al. [3] estimated that the potential for reducing voltage losses by eliminating the triplet recombination pathway is around 60 mV, which is small compared to the overall voltage losses of ~900 mV and ~700 mV for our ZnPc:C₆₀ and F₄ZnPc:C₆₀ blends. These voltage loss values are typical for fullerene blends with large energetic offsets, where factors such as the S1 – CT offset dominate.

Another recombination channel that is often assumed to dominate voltage losses is vibration-induced direct recombination from the CT state to the GS. This loss decreases for increasing CT state energies according to the energy gap law [1], which is consistent with our observed trends. We appreciate that this could have been discussed better and have now expanded our discussion of losses in the Supplementary Information. We have also added additional modelling following equations 1 & 2 of Gillet et al. [3] to better describe the interplay between increased non-radiative recombination via triplet states and reduced vibrational coupling with increasing CT state energies. The newly added figure is provided below for convenience.

Figure S12. Dependence of energy losses on charge transfer state energies (E_{CT} ; top) and radiative recombination events (bottom). a) Non-radiative energy losses ($q\Delta V_{oc}^{nr}$) and b) radiative energy losses ($q\Delta V_{oc}^{rad}$) decrease with increasing E_{CT} . c) $q\Delta V_{oc}^{nr}$ also depends on the fraction of radiative

recombination events (χ) and photoluminescence quantum efficiency (ϕ_{PL}), modelled after equations 1 and 2 in Gillett et al.[3] We gradually increased ϕ_{PL} from 0.4% to 2.0%, which is reasonable for organic solar cells. We also assume a photon out-coupling efficiency of 0.3, a charge balance factor of 1, and a temperature of 300 K.[3] When χ increases, $q\Delta V_{oc}^{nr}$ decreases. Similarly, when ϕ_{PL} increases, $q\Delta V_{oc}^{nr}$ is reduced. The latter can be achieved, for instance, by increasing E_{CT} and reducing non-radiative coupling between the CT and ground state (according to the energy gap law [1]). As a result, the total $q\Delta V_{oc}^{nr}$ measured for different material systems arises from the interplay between these different factors.

2) As also noted by Ref. 3, I still find the constant delta ECS a bit unsatisfactory, and the values are negative and very high (-0.5 eV). I'm not sure how efficient the CT dissociation can be for such a negative and high energy barrier as in ZnPc:C60. Also, are the authors implying the same rate constant of CS dissociation (not the yield) for all blends since delta ECS is the same? This would need further justification as well. My feeling is that if the authors want to make a strong statement about CS dissociation, the Delta ECS should be properly investigated as it directly affects the dissociation yield. Clarifying this point could make the conclusions more convincing.

Response: We would like to highlight that the focus of our calculations was on estimating E_{CT} , and not on drawing conclusions about the energetic difference between CS and CT states. We used three approaches to estimate E_{CT} : (i) EQE fitting, (ii) TDDFT calculations and (iii) the “electrostatic” approach referred to by the reviewer. In the latter case, we use a simplified model, where we estimate $E_{CT} = E_{CS} + E_{el}$.

Our DFT geometry optimizations performed on donor-acceptor complexes indicate that the donor-acceptor distance is nearly the same for all studied systems. Therefore, in accordance with the computed distances, we took a value of 0.5 eV for the electrostatic energy (E_{el}) for all blends. A more reliable approach to determine E_{el} would be to perform MD simulations of the blends, as we have done e.g. for P3HT:PCBM [4], and to compute the distribution of electrostatic energies. Note that in the case of P3HT:PCBM, we found an average E_{el} of about 0.56 eV. Unfortunately, the MD simulations for dilute blends are computationally expensive as one would have to consider very large systems (to reproduce small the donor content).

Based on this, the electrostatic approach suggests that ΔE_{CS} is indeed the same for all blends. However, since this model is based on very rough approximations, we moved the discussion of the resulting CT state energies to the Supplementary Information, and instead decided to focus on the TDDFT results in the paper. We modified the main text and the Supplementary Information, to make the issues surrounding different calculation approaches clearer. If the CT state energies derived by the first two approaches are used to calculate ΔE_{CS} , the energetic difference between the CT and CS state indeed increases as a function of donor fluorination.

For the calculations of E_{CS} , we used the medium polarization model that does not account for quadrupole moments. Therefore, the estimated E_{CS} are likely overestimated. Moreover, for the charge separation process, free energy matters. Due to the entropy contributions, the free energy in the blends could be significantly smaller than the energy of the CS state. Clearly, additional investigations are needed to better quantify both CS state energies and the free energy of charge separation in OPVs. This issue, however, is beyond the scope of our current investigation.

Reviewer #2 (Remarks to the Author):

I really appreciate the effort the authors have made to address the concerns of the reviewers. The additional data provided and their analysis adds substantially in supporting their conclusions, and they clearly improved their manuscript in the revised version. However, I think my second major criticism was only partially addressed. This point was related to how much the findings could be transferred to BHJ systems in general. This is of course a very difficult task to achieve. Overall, I would recommend this manuscript now for publication in Nature Communications.

Response: We thank the reviewer for their positive evaluation of our revised manuscript.

Reviewer #3 (Remarks to the Author):

The authors have done a very good job in revising the manuscript. I am happy with this version.

Response: We thank the reviewer for their positive evaluation of our revised manuscript.

References

- [1] J. Benduhn *et al.*, “Intrinsic non-radiative voltage losses in fullerene-based organic solar cells,” *Nat Energy*, vol. 2, no. 6, 2017, doi: 10.1038/nenergy.2017.53.
- [2] X. K. Chen and J. L. Brédas, “Voltage Losses in Organic Solar Cells: Understanding the Contributions of Intramolecular Vibrations to Nonradiative Recombinations,” *Adv Energy Mater*, vol. 8, no. 9, Mar. 2018, doi: 10.1002/aenm.201702227.
- [3] A. J. Gillett *et al.*, “The role of charge recombination to triplet excitons in organic solar cells,” *Nature*, vol. 597, no. 7878, pp. 666–671, Sep. 2021, doi: 10.1038/s41586-021-03840-5.
- [4] Z. Zheng, N. R. Tummala, T. Wang, V. Coropceanu, and J. L. Brédas, “Charge-Transfer States at Organic–Organic Interfaces: Impact of Static and Dynamic Disorders,” *Adv Energy Mater*, vol. 9, no. 14, Apr. 2019, doi: 10.1002/aenm.201803926.

REVIEWERS' COMMENTS

Reviewer #1 (Remarks to the Author):

I appreciate the authors took my comments very seriously, and the revised version is clearly improved, therefore, I believe this manuscript is ready for publication.